# Preliminary Evaluation of Plasma circ_0009910, circ_0027478, and miR-1236-3p as Diagnostic and Prognostic Biomarkers in Hepatocellular Carcinoma

**DOI:** 10.3390/ijms26104842

**Published:** 2025-05-19

**Authors:** Mona Samy Awed, Abeer Ibrahim, Omnia Ezzat, Amal Fawzy, Deema Kamal Sabir, Abdullah F. Radwan

**Affiliations:** 1Department of Biochemistry, Faculty of Pharmacy, Egyptian Russian University, Cairo 11829, Egypt; mona-sami@eru.edu.eg (M.S.A.); omniaezzat@yahoo.com (O.E.); 2Department of Biochemistry and Molecular Biology, Faculty of Pharmacy (Girls), Al-Azhar University, Cairo 11651, Egypt; abeeribrahim75@yahoo.com; 3Department of Clinical Pathology, National Cancer Institute, Cairo University, Cairo 11796, Egypt; 4Department of medical surgical Nursing, College of Nursing, Princess Nourah bint Abdulrahman University, P.O. Box 84428, Riyadh 11671, Saudi Arabia; 5College of Pharmacy, University of Kut, Wasit 52001, Iraq

**Keywords:** hepatocellular carcinoma (HCC), circRNAs, circ_0009910, circ_0027478, miR-1236-3, plasma biomarkers

## Abstract

Circular RNAs (circRNAs) are increasingly recognized as significant regulators in multiple cancers, such as hepatocellular carcinoma (HCC), frequently affecting microRNA (miRNA) expression. The diagnostic and prognostic roles of circRNAs, specifically *circ_0009910* and *circ_0027478*, in conjunction with miR-1236-3p, in HCC, have not yet been fully investigated. In this pilot study, we assessed the expression levels of circ_0009910, circ_0027478, and miR-1236-3p in plasma samples from 100 patients diagnosed with HCC and 50 healthy controls through reverse transcriptase–quantitative polymerase chain reaction (RT-qPCR). The diagnostic performance was evaluated using receiver operating characteristic (ROC) curve analysis, and correlations with clinicopathological features were examined. Circ_0009910 and circ_0027478 exhibited significant upregulation in patients with HCC (*p* < 0.05), whereas miR-1236-3p demonstrated downregulation (*p* < 0.05). Circ_0009910 demonstrated significant diagnostic accuracy (area under the curve [AUC] = 0.90), effectively differentiating HCC from controls and showing a correlation with tumor size, metastasis, and alpha-fetoprotein (AFP) levels (*p* < 0.05). Both circ_0009910 and circ_0027478 exhibited a positive correlation with clinicopathological features, whereas miR-1236-3p demonstrated an inverse correlation. Logistic regression validated the diagnostic and prognostic capabilities of these biomarkers. The results indicate that circ_0009910, circ_0027478, and miR-1236-3p, in conjunction with AFP three, present a promising diagnostic and prognostic profile for HCC. Additional validation in larger cohorts is required to establish their clinical utility.

## 1. Introduction

Among the most prevalent forms of cancer worldwide is cellular carcinoma of the liver, which is the primary cancer of hepatocytes. In addition, it ranks fifth among cancers in men and eighth among cancers in women, making it the second most common cause of cancer-related deaths [1]. The fact that HCC is typically discovered in its later stages is the main cause of the disease’s continued rise in death rates [2]. Additionally, no treatment may provide a cure for advanced HCC [3]. With a global prevalence of over 900,000 new cases per year, it is the most common primary liver cancer and the third leading cause of cancer-related deaths globally [4]. Its incidence is highest in East Asia, sub-Saharan Africa, and parts of Europe, primarily due to chronic hepatitis B and C infections, alcohol-related liver disease, and non-alcoholic fatty liver disease (NAFLD) [5]. Despite advancements in diagnosis, current biomarkers, such as alpha-fetoprotein (AFP), have limitations, including low sensitivity and specificity, especially when it comes to detecting early-stage HCC. Emerging biomarkers, such as circulating tumor DNA (ctDNA), exosomal RNAs, and non-coding RNAs, show promise but lack standardized validation for widespread clinical use. Therefore, combining imaging techniques with multi-omics biomarker panels is being investigated to improve early detection and prognosis [6].

Early identification of HCC has traditionally depended on the use of ultrasonography (US) and serological evaluations of AFP, being the most extensively researched and often used biomarker for HCC detection [7]. Nevertheless, the accuracy and effectiveness of using US/AFP for early detection of a substantial fraction of early-onset HCC remain inadequate [8]. Recently, there has been a lot of interest in identifying biomarkers in physiological fluids to help detect HCC early. However, none of these biological biomarkers are useful for therapeutic usage and can attain a sensitivity and specificity of >90% [9,10]. Periodic HCC screening is recommended for at-risk people who often have underlying liver cirrhosis and/or persistent viral infection of the hepatitis virus [11].

Researchers are currently studying the function of non-coding RNA (ncRNA) molecules in HCC progression and development, with the goal of uncovering new therapeutic targets and strategies [12]. The recent use of next-generation sequencing (NGS) techniques has made it possible to analyze tumor mutational profiles based on different types of circulating tumor biomarkers [13]. One of the ncRNAs essential to many biological functions is circular RNA (circRNA), which is especially useful for controlling gene expression [14]. CircRNAs are single-stranded, covalently closed RNA molecules that are mostly generated by reverse back-splicing pre-mRNA [15]. CircRNAs have been linked to many phases of tumor growth, including metastasis, proliferation, apoptosis, and differentiation [16]. Circ_0009910 and circ_0027478 are exonic circRNAs formed through a back-splicing mechanism, where a downstream exon is covalently linked to an upstream exon to form a stable, closed-loop structure. circ_0009910 originates from the *mitofusin 2* (MFN2) gene on chromosome 1, while circ_0027478 is derived from the *nucleoporin 107* (NUP107) gene on chromosome 12. These circular RNAs lack 5′ and 3′ ends, making them resistant to exonuclease-mediated degradation and contributing to their stability in circulation. Bioinformatic secondary structure analyses suggest that their looped configurations may serve as binding platforms for RNA-binding proteins and miRNAs, indicating a potential regulatory role in gene expression relevant to tumor biology (Appendix A). Numerous studies have looked at how circRNAs contribute to the growth of HCC [17,18,19]. However, not much is understood about the specific functions and underlying processes of circRNAs in HCC.

circRNAs use three primary methods to carry out their biological activities: translating into proteins, communicating with RNA binding proteins (RBPs), and serving as microRNA (miRNA) sponges that are short lengths of 22 nucleotides, which can either inhibit or cause cancer [20,21]. Regarding the most important and particular mechanism through which a circRNA operates may contribute to its final biological role, a growing body of research recognizes that circRNAs act as competitive endogenous RNAs (ceRNAs) and may affect the production of other miRNAs [22,23]; that is, a small ncRNA molecule helps regulate gene expression by binding to mRNA, preventing it from being translated into protein. miRNAs play a key role in controlling various cellular processes, such as growth, development, and disease [24]. They have shown the ability to interact with the 3′-untranslated regions (UTRs) of mRNA to alter its levels [25].

miR-1236-3p, a significant malignancy suppressor in various cancers, including HCC [26,27], plays a pivotal role in inhibiting cancer cell division, invasion, and migration. It achieves this by targeting specific oncogenes and pathways. For instance, miR-1236-3p has been shown to suppress the epithelial–mesenchymal transition (EMT), a substantial mechanism in cancer spreading [28]. In HCC, miR-1236-3p diminishes the PI3K/Akt route by downregulating phosphatase and tensin homolog (PTEN) and AFP, thereby slowing the growth and spread of the tumor [29].

It was determined using bioinformatics tools that circ_0027478 interacts with miR-1236-3p, hence decreasing the availability of miR-1236-3p to target oncogenic pathways. This interaction may promote the advancement of tumors by downregulating miR-1236-3p’s ability to suppress oncogenes like AFP and the PI3K/Akt route, highlighting the possible therapeutic implications of targeting the circ_0027478/miR-1236-3p axis.

As far as we know, there is a paucity of literature on circ-0009910 and circ-0027478 as markers in HCC, though they are known to contribute to HCC carcinogenesis. These circRNAs have been implicated in other tumors such as gastric cancer [30], ovarian malignancy [31], osteosarcoma [32], and leukemia [33]. However, for the first time, these biomarkers are being assessed in the plasma of patients with HCC, offering a potential breakthrough in HCC research. Further research is desperately needed to strengthen their roles as prospective therapeutic targets and diagnostic tools in HCC.

Building on previous findings, this study aims to investigate the plasma expression of circ-0009910 and circ_0027478 in patients with HCC, making them the first to be assessed in plasma for HCC. This study also examines the link between miR-1236-3p and HCC, specifically in relation to circ_0027478. Additionally, it explores the relationship that exists between the expression levels of these markers and clinicopathological parameters, such as tumor size, stage, and spreading of the cancer. Finally, the prognostic and diagnostic performance of circ_0009910, circ_0027478, and miR-1236-3p is compared with the conventional AFP marker in patients with HCC.

## 2. Results

### 2.1. Demographic and Clinicopathological Characteristics and Laboratory Results of the Studied Groups

Table 1 presents a comprehensive overview of the demographic, laboratory, and pathological characteristics of the studied groups under investigation. The HCC and control groups were well matched for age (*p* = 0.22) and gender (*p* = 0.64), ensuring that differences in biomarker expression were not influenced by demographic variability. However, a higher proportion of males in both the HCC and control groups, accounting for 58% and 54%, respectively, was noted.

Concerning the laboratory examinations, a significant disparity was seen among the groups under study in terms of aspartate transaminase (AST), alanine transaminase (ALT), AFP levels, the presence of hepatitis C virus (HCV) antibodies, international normalized ratio (INR), prothrombin time (PT), albumin, and bilirubin with a statistically significant *p*-value of 0.001. The values of these laboratory parameters were taken from the patients’ medical records, as provided by the hospital’s laboratory system.

The reported instances of HCC were distributed as follows: 44% were seen in the left lobe of the liver, 35% were discovered in the right lobe, and the remaining cases, accounting for 21%, were identified in both the right and left lobes. Furthermore, the size of the tumors has a range of 1 cm to 9.5 cm. Metastatic HCC was seen in just 25% of the individuals. Moreover, it has been shown that 49% of patients with HCC have received a diagnosis of Barcelona Clinic Liver Cancer (BCLC) in the early stages (A and B). Conversely, 51% of the patients with HCC identified with BCLC have advanced to late stages (C and D).

### 2.2. The Plasma Expression Distribution of has_circ_0009910, hsa_circ_0027478, and miR-1236-3p

When comparing patients with HCC to the control group, the plasma expression levels of circ_0009910 and circ_0027478 showed significant upregulation, with median fold increases of 35.6 (*p* < 0.001) and 3.8 (*p* < 0.001), respectively. miR-1236-3p, on the other hand, was substantially downregulated, with a median fold change of 0.13 (*p* < 0.001), as shown in Table 2.

A comparative assessment was conducted to evaluate the levels of the relative expression of circ_0009910 and circ_0027478 in patients with HCC who were categorized as early-stage (*n* = 53) and those identified as late-stage (*n* = 47), using the TNM staging method. The findings of the research denote that the late-stage HCC group had substantially elevated levels of circ_0009910 and circ_0027478 expression in comparison to the early-stage HCC group (*p* < 0.0001 and *p* = 0.004), respectively, as seen in Figure 1. However, there was no statistically noteworthy change seen in the expression levels of miR-1236-3p in relation to TNM staging. Furthermore, while categorizing the patients with HCC based on the BCLC staging method, it was seen that 28% of the patients were categorized as Stage A (early stage), 21% as Stage B (intermediate stage), 34% as Stage C (advanced stage), and 17% as Stage D (severe liver damage). The analysis revealed that circ_0009910 had the ability to distinguish between almost all four phases, with the exception of circ_0027478, which demonstrated differentiation just between Stage A and Stage D only (Figure 1).

With respect to the quantity of tumor lesions, patients with HCC may be categorized into two groups: single (*n* = 32) and multiple (*n* = 68). The findings of the investigation indicated that the expression levels of circ_0009910 and circ_0027478 exhibited the capacity to differentiate between the two groups, with statistically significant differences of 0.0001 and 0.0086, respectively (Figure 2). When comparing the relative expression levels of circ_0009910 within the HCC group between the patients who had metastasis and those who had not, it was found that the metastatic HCC group had higher circ_0009910 and circ_0027478 expression levels than the non-metastatic HCC group (*p* < 0.0001) and (*p* < 0.005), respectively, as shown in Figure 2.

Patients with HCC are categorized into four distinct groups based on the presence or absence of cirrhosis. These categories were as follows: Group 0 (*n* = 16) exhibited no fibrosis, Group 1 (*n* = 39) displayed mild fibrosis, Group 2 (*n* = 28) displayed moderate fibrosis, and Group 3 (*n* = 17) exhibited severe fibrosis or cirrhosis. The expression levels of plasma hsa_circ_0009910 were found to possess the ability to distinguish between all subgroups, with the exception of moderate and severe fibrosis, as depicted in Figure 3. Conversely, the plasma expression levels of circ_0027478 were observed to differentiate between patients with HCC with severe cirrhosis and those with grades 0, 1, and 2, with *p*-values of 0.0105, 0.0137, and 0.0327, respectively (Figure 3).

In addition, the patients with HCC were classified into two groups based on the size of their tumor lesions. The first group consisted of 40 patients with tumor lesions smaller than 3 cm, while the second group consisted of 60 patients with HCC lesions larger than 3 cm. By analyzing the plasma expression levels of circ_0009910 and circ_0027478, significant differences were observed between the two subgroups, with *p*-values of 0.0001 and 0.006, respectively, as shown in Figure 3.

Finally, there were no significant differences seen in the plasma expression levels of circ_0009910, circ_0027478, and miR-1236-3p between the subgroups classified based on Okuda staging and child staging.

### 2.3. Plasma Levels of AFP in the Studied Groups

Regarding the relationship between AFP levels and various clinical parameters in patients with HCC, Figure 4 shows that AFP levels are significantly elevated in patients with HCC compared to controls (*p* < 0.0001). In addition, it demonstrates a progressive increase in AFP levels with advancing cirrhosis grades (0–3), with significant differences observed between some groups such as between cirrhosis grade 0 vs. 2 with *p* = 0.0092; grade 1 vs. 2 with *p* = 0.0294; grade 0 vs. 3 with *p* = 0.0029; and grade 1 vs. 3. with *p* = 0.0094. Panel C highlights elevated AFP levels across the different BCLC stages, with statistically significant differences noted between certain stages. Panel D indicates that patients with metastasis have significantly higher AFP levels than non-metastatic patients (*p* = 0.0232). Panel E reveals that AFP levels are substantially higher in late-stage HCC compared to early-stage cases (*p* = 0.0013). Finally, Panel F shows a significant increase in AFP levels in tumors larger than 3 cm compared to those smaller than 3 cm (*p* = 0.0205).

### 2.4. Comparative Diagnostic Accuracy of the Studied Parameters

Concerning circular RNAs, ROC analysis demonstrated that plasma expression levels of circ_0009910 and circ_0027478 effectively differentiated patients with HCC from healthy controls, yielding an area under the curve (AUC) of 0.90 (95% confidence interval [CI] = 0.98 to 1.000, *p* < 0.0001), with a sensitivity of 96% and specificity of 100% at a cutoff > 4.8-fold. Additionally, an AUC of 0.80 (95% CI = 0.823 to 0.934, *p* < 0.0001) was observed, with a sensitivity of 81% and specificity of 82% at a cutoff > 1.76-fold, respectively (Figure 5A,B).

In the instance of miR-1236, its plasma expression level was shown to distinguish patients with HCC from healthy controls, achieving an area under the AUC of 0.90 (95% CI = 0.9 to 0.975, *p* < 0.0001), with a sensitivity of 82% and specificity of 91% at a cutoff less than 0.4-fold (Figure 5C).

Regarding plasma AFP, ROC analysis demonstrated its ability to differentiate HCC from healthy controls, with an AUC of 0.83 (95% CI: 0.815 to 0.925, *p* < 0.0001), with a sensitivity of 74% and specificity of 94% at a cutoff > 27.5 ng/mL (Figure 5D).

### 2.5. Enhancing the Diagnostic Performance of AFP Through Integration with the Studied circRNAs and miR-1236

To improve the diagnostic accuracy for HCC, multiple biomarker combinations were evaluated using logistic regression models, with performance assessed via ROC curve analysis. The integrated models demonstrated superior diagnostic efficacy compared to individual biomarkers, underscoring the potential of multi-analyte approaches in HCC detection.

As illustrated in Figure 6, Model 1, comprising AFP and circ_0027478, yielded an area under the ROC AUC of 0.93 (95% CI: 0.893–0.971, *p* < 0.0001), with a sensitivity of 86% and specificity of 90%. Model 2, integrating AFP with miR-1236-3p, achieved an AUC of 0.97 (95% CI: 0.952–0.995, *p* < 0.0001), with 88% sensitivity and 95% specificity.

In Model 3, the combination of circ_0027478 and circ_0009910 resulted in an AUC of 0.996 (95% CI: 0.990–1.000, *p* < 0.0001), with 97% sensitivity and 100% specificity. Model 4, integrating miR-1236-3p and circ_0009910, demonstrated the highest diagnostic performance, with an AUC of 0.999 (95% CI: 0.997–1.000, *p* < 0.0001), 99% sensitivity, and 100% specificity. These findings highlight a marked improvement in diagnostic capability when AFP is combined with specific non-coding RNAs, increasing its AUC from 0.83 as a standalone marker to 0.93 and 0.97 when paired with circ_0027478 and miR-1236-3p, respectively.

### 2.6. Prognostic Significance of the Studied Biomarkers

circ_0009910 and circ_0027478 plasma expression levels also distinguished between patients with metastatic HCC and those without: circ_0009910 shows an AUC = 0.77, 95% CI = 0.671 to 0.882, *p* < 0.0001, 64% sensitivity, and 81% specificity with a cutoff > 50.8-fold, while circ_0027478 displays an AUC = 0.72, 95% CI = 0.612 to 0.843, *p* = 0.001, 76% sensitivity, and 66% specificity with a cutoff > 4-fold, respectively, as shown in Figure 7A,B.

In contrast, miR-1236-3p exhibited no discriminatory power in this context, with an AUC = 0.5, *p* = 0.5, a specificity of 65%, sensitivity of 48.5%, and a cutoff < 0.09, as revealed in Figure 7C. Meanwhile, AFP demonstrated moderate discriminatory ability with an AUC = 0.7, *p* = 0.002, sensitivity of 72%, and specificity 66.7% (Figure 7D).

### 2.7. Analysis Using Logistic Regression

Univariate and multivariate logistic regression analyses were conducted to identify the predictors related to HCC risk (Table 3). circ_0009910, circ_0027478, miRNA-1236, and AFP expression levels were identified as significant predictors correlated with the risk of being diagnosed with HCC in the univariate testing (*p* < 0.05). In sequential forward multivariate testing, all indicators emerged as non-significant predictors of the chance of HCC diagnosis (*p* >0.05).

### 2.8. Association Between Expression Levels of Analyzed Parameters and Clinicopathological Features in Patients with HCC

A significant inverse association was observed between hsa_circ_0027478 and miRNA1236-3p in the HCC group. Significant positive correlations were observed between circ_0009910 and circ_0027478 in the HCC group (r = 0.325, *p* = 0.001). Circular RNAs 0009910 and 0027478 exhibit significant positive correlations with tumor size and AFP levels (*p* < 0.001). Positive correlations were observed between circRNAs 0009910 and 0027478 and various clinicopathological parameters in patients with HCC, including tumor stage as per BCLC and TNM staging systems, distant metastasis, and cirrhosis. **AFP** levels also correlate positively with key HCC indicators such as **tumor size** (r = 0.366, *p* < 0.0001), **TNM** (r = 0.494, *p* < 0.0001), and **BCLC** (r = 0.554, *p* < 0.0001) (Figure 8 and Table 4).

## 3. Discussion

The predominant form of liver cancer, HCC, constitutes roughly 80% of all total hepatic neoplasms and is among the foremost reasons of mortality worldwide [35]. Together with cancer recurrence and deadly metastasis, the intractable tumor microenvironment is one of the three main unresolved difficulties in therapeutic management, having a substantial impact on the progression and development of HCC [36].

Recent advancements in genome-wide transcriptomic profiling and bioinformatics have enabled the systematic elucidation of the molecular regulatory processes linked to cancer and development. Nonetheless, substantial sample numbers are necessary to detect circRNA expression variations with enhanced statistical power [37]. Over the past few years, circRNAs have become apparent as a significant category of ncRNA fragments, many of these are pivotal in the initiation and progression of malignancy via several routes. They are often manifested in specific to tissue and cancer-specific manners, and with the proliferation of research data, it is progressively uncovering that these compounds could possess substantial therapeutic significance and application chances [38,39]. circRNAs modulate cellular processes and illness, especially cancer [40]. Recent data indicate the function of circRNAs in tumor progression, suggesting their potential as new biomarkers and therapeutic targets for cancer therapy [41].

This study identified two increased circRNAs, circ_0009910 and circ_0027478, by examining two tissue microarray datasets, indicating that these unregulated circRNAs in the tissue of HCC may be released into plasma [42,43]. Utilizing the CircInteractome database and CircBank to anticipate miRNA response elements (MREs) or their binding sites, miR-1236-3p was identified in both databases and regarded as a probable target of circ_0027478. Cumulative evidence suggests that miR-1236-3p acts as a tumor suppressor in the proliferation, migration, and invasion of liver cancer via modulating the PTEN/PI3K/Akt pathway mediated by AFP. The cumulative data demonstrate that miR-1236-3p works as a tumor suppressor in the growth, proliferation, invasion and migration of liver cancer by modulating the PTEN/PI3K/Akt route that is mediated by AFP [29,44].

However, some researchers have identified the transcription profile of circ_0009910 in tissue of six cancer types, cancer of the ovaries, breast cancer, gastric cancer, HCC, osteosarcoma, and acute myeloid leukemia (AML), by high-throughput sequencing of various cancer tissues [30,31,32,45,46,47]. In addition, circ_0027478 was assessed once in HCC tissues [48]. For the first time, circ_0009910 is to be assessed as a novel non-invasive diagnostic and prognostic non-invasive plasma indicator in HCC.

The upregulation of circ_0009910 and circ_0027478 in the plasma of HCC individuals, as well as the downregulation of miR-1236-3p, are promising findings that could potentially revolutionize the research hypothesis in HCC. The overexpression of circ_0009910 and circ_0027478 in the plasma of HCC individuals aligns with their upregulated expression in HCC tissue, as observed in another study, suggesting its oncogenic role in this type of cancer [48,49]. Like some other circRNAs, circ_0009910 and circ_0027478 could effectively differentiate between subgroups based on tumor differentiation, size, metastasis, cirrhosis grading, and TNM staging that matches the results of other studies [48,50,51], offering new avenues for personalized treatment and patient care.

As mentioned by Zhang et al., the abnormal expression profile of circRNAs is potentially utilized for the characterization of HCC [52]. Within the BCLC staging system for HCC, the current research has revealed that circ_0009910 exhibits enhanced diagnostic efficacy relative to circ_0027478. Specifically, circ_0009910 may distinguish among BCLC Stages A to D and the TNM staging system, offering a more sophisticated evaluation of illness development. In contrast, hsa_circ_0027478 can differentiate between late-stage (C and D) and early phases (A and B). This underscores circ_0009910 as a more sensitive biomarker, providing enhanced staging precision and distinction based on disease severity. Suggesting promising possibilities for future research in HCC diagnosis and treatment. Furthermore, both circRNAs can differentiate between patients with HCC with malignancies smaller than 3 cm and those with tumors measuring 3 cm or larger suggesting the concomitant association between the investigated circRNAs and HCC progression.

According to Wang et al., there are accumulating data that suggest that changed circRNA expression might influence the carcinogenesis and development of HCC. Additionally, these circRNAs have tremendous promise in hepatic cancer diagnosis, therapy, and prognosis [53]. The studied circ_0009910 and circ_0027478 can distinguish between patients with HCC with a single tumor lesion and those with numerous tumor lesions. On the other hand, circ_0009910 and circ_0027478 can proficiently distinguish between metastatic HCC and non-metastatic HCC. Their ability to differentiate between metastatic and non-metastatic types further validates their usefulness in the clinical assessment of HCC development. Moreover, in the context of cirrhosis, circ_0009910 exhibits enhanced differentiation capabilities. It can separately differentiate between non-cirrhotic HCC and various stages of cirrhosis, including fibrosis and mild, moderate, and severe cirrhosis, as well as between fibrosis vs. moderate and severe cirrhosis, separately. On the other hand, the intriguing circ_0027478 plays a crucial role in distinguishing between non-cirrhotic HCC vs. severe cirrhotic HCC and fibrotic HCC vs. severe cirrhotic HCC only. circ_0009910 serves as the optimal marker for exhibiting different levels of cirrhosis.

The diagnostic efficacy of the examined markers indicates that circulating circRNAs and miRNAs serve as accessible, valid, and precise genetic testing across many cancer types [41,54]. The current study revealed that plasma circ_0009910, circ_0027478, and miRNA-1236-3p were uniquely expressed in individuals with HCC compared to healthy controls, and they effectively differentiated HCC from control groups with high specificity and sensitivity, suggesting that plasma circ_0009910, circ_0027478, and miRNA-1236 could function as potential novel biomarkers for the screening and diagnosis of HCC. Moreover, this study assessed their potential diagnostic and prognostic significance. Notably, circ_0009910 and miR-1236-3p demonstrated greater accuracy, enhanced sensitivity, and specificity compared to circ_0027478 and AFP. These results identify plasma circ_0009910 and miR-1236-3p as dependable, harmless, early biomarkers and a prospective therapeutic target for HCC therapy.

Although AFP remains a common biomarker in HCC surveillance, its diagnostic performance is suboptimal. Levels can be elevated in benign liver conditions such as hepatitis and cirrhosis, reducing specificity [55], while many early stage HCC cases present with normal AFP levels, limiting sensitivity [56]. With sensitivity ranging from 39–65% at conventional cutoffs, AFP alone is inadequate for reliable early detection. Furthermore, its levels do not consistently reflect tumor burden or progression [57]. These limitations emphasize the need for more accurate biomarkers. Our findings suggest that combining AFP with the studied ncRNAs may improve early HCC detection. However, beyond statistical performance, clinical applicability must be evaluated, as the utility of any diagnostic tool depends on balancing sensitivity and specificity according to the intended clinical context. The combined ROC curves support this potential enhancement and align with accumulating evidence that circRNAs and miRNAs play crucial roles in HCC pathogenesis through their involvement in tumor-specific regulatory networks, including cell proliferation, invasion, and immune evasion [58,59].

Referring to the prognostic accuracy, circ_0009910 differentiated between individuals with metastatic HCC and those without, exhibiting superior sensitivity and specificity compared to circ_0027478, miR-1236-3p, and AFP. This supported the notion referring to the predictive capacities of circRNAs in HCC, such as circ_101368, which is considerably enhanced in HCC tissue and is connected with a worse prognosis in HCC individuals [60]. However, miR-1236-3p had no discriminatory power in the metastatic context, while AFP showed moderate prognostic ability, which is consistent with previous results. For instance, a randomized clinical study demonstrated that preoperative AFP levels are not linked with postoperative survival in HCC individuals, likely due to tumor stage heterogeneity [61]. Similarly, another study revealed that AFP levels lack prognostic significance in patients with HCC with small cancer size (≤3 cm in diameter) who underwent curative treatments, including liver transplantation, liver resection, or radiofrequency thermal ablation [61]. The variability across these curative modalities suggests that further investigation is warranted to clarify these associations.

Similar to other carcinogenic circRNAs, such as circβ-catenin, which originates from the host’s β-catenin and is significantly expressed in HCC, circRNAs have a significant role in hepatocarcinogenesis. Circβ-catenin promotes HCC progression by activating the Wnt/β-catenin signaling cascade [62]. Likewise, circ_0000105 is upregulated in HCC, facilitating cell proliferation and inhibiting apoptosis through the miR-498/PIK3R1 route [63]. In another study, Niu et al. reported that circ_0091579 is significantly upregulated in hepatic cancer and promotes cell proliferation and metastasis by downregulating miRNA-490-3p [64]. In our study, circ_0009910, an exonic circRNA found on chromosome 1 at positions 12049222-12052747 and hosted by the mitofusin 2 gene, was found to be elevated in patients with HCC. This elevation was associated with increased risk and poor prognosis. Similarly, circ_0027478, another exonic circRNA, located on chromosome 12 at positions 69109407-69125499 and hosted by the nucleoporin 107 gene, exhibited oncogenic properties. We explored the potential mechanism of circ_0027478 using bioinformatics tools and identified its cytoplasmic localization, suggesting that it might exert its effects by sequestering miRNAs. Bioinformatic analysis revealed that circ_0027478 interacts with miR-1236, further supporting its regulatory role in HCC progression.

Growing data consistently indicate that miR-1236-3p is involved in cancer cell growth, proliferation, migration, invasion, and apoptosis. Additionally, miR-1236 is associated with EMT, a crucial marker of the metastatic phase. Furthermore, miR-1236-3p is modulated by many recently identified lncRNAs and circRNAs [65]. Prior research has shown that miR-1236-3p has inhibitory effects across various tumors [29,66,67,68]. In glioblastoma, circ_0074362 was closely linked to miR-1236-3p, which targets the gene HOXB7, resulting in glioma cell advancement [69,70]. circ_0027446 facilitates the spreading and EMT of lung cancer via the miR-1236-3p/ZEB1 pathway [68]. circRAPGEF5 enhances the progression of lung adenocarcinoma via the miR-1236-3p/ZEB1 pathway [71]. Thus, the potential contribution of the circ_0027478/miR-1236-3p tract in the progression of HCC has been suggested. However, this hypothesis requires further extensive research to validate its mechanistic role and clinical significance.

AFP is an oncofetal polypeptide that is primarily generated by the fetus liver and yolk sac [72]. In individuals, AFP is either undetected or exists in tiny quantities (<20 ng/mL); however, it is re-expressed in over 70% of HCC cases [73]. As previously established, AFP is widely used as a biomarker for monitoring HCC in clinical environments [74]. This aligns with the current study’s findings, which show that elevated AFP levels in the blood are linked with larger tumor size, higher rates of spreading of malignancy, and more advanced cirrhosis. These results suggest that AFP correlates with disease progression and severity in HCC. Additionally, AFP demonstrates high specificity and intermediate sensitivity as a diagnostic marker for HCC, further supporting its utility in clinical diagnosis and its potential oncogenic role in HCC progression.

Bioinformatics analysis identified an interaction between circ_0027478 and miR-1236-3p in HCC. Previous research has demonstrated that miR-1236-3p directly targets the 3′ untranslated region (UTR) of AFP in liver cancer, leading to its downregulation. Functionally, miR-1236-3p suppresses, while AFP promotes the proliferation, migration, and invasion of HCC cells. Furthermore, AFP overexpression counteracts the inhibitory effects of miR-1236-3p. Mechanistically, AFP was found to enhance PTEN ubiquitination, thereby reducing PTEN levels, whereas miR-1236-3p inhibits the PI3K/Akt signaling pathway [29]. This indicates that circ_0027478/miR-1236-3p/AFP could be identified as a therapeutic target in patients with HCC.

Eventually, this study identified significant correlations between the circRNAs circ_0009910, circ_0027478, and miRNA-1236-3p and key clinicopathological features in HCC, agreeing with the general idea that came in the research that studied circRNA signature in HCC [42]. An inverse association between hsa_circ_0027478 and miRNA-1236-3p suggests that circ_0027478 can act as a miRNA sequester, promoting HCC advancement by inhibiting miRNA-1236-3p’s tumor-suppressive effects, matching the results of the research that studied the role of some circRNAs, miRNAs, and their interactions in the development and progress of HCC [75]. Both circ_0009910 and circ_0027478 significantly and positively correlated with tumor size, AFP levels, and disease severity, including advanced tumor stage, metastasis, and cirrhosis, highlighting their potential as biomarkers for HCC progression. These findings suggest that circ_0009910 and circ_0027478 play key roles in HCC development and could serve as valuable prognostic markers or therapeutic targets, though further research is needed to confirm their specific roles.

circ_0027478 shows significant positive correlations with AFP, Size, TNM, and BCLC (all with *p*-values < 0.0001), suggesting that this circRNA might be involved in pathways related to tumor growth, metastasis, and HCC progression. Specifically, the correlation with AFP suggests a potential role in regulating AFP expression, a well-established diagnostic marker for HCC. circ_0009910 also correlates significantly with AFP, Size, TNM, and BCLC, indicating that it might be implicated in similar tumor progression pathways. Its strong correlation with TNM and size suggests a role in tumor staging, further supporting its potential as a prognostic biomarker. miR-1236-3p appears to have negative correlations with several clinical parameters, such as circ_0027478 and AFP. This suggests that miR-1236-3p may function as a tumor suppressor, potentially inhibiting the expression of key genes involved in HCC progression. These correlations suggest that circ_0009910 and circ_0027478 may influence key biological processes like tumor growth, angiogenesis, and metastasis, while miR-1236-3p could regulate tumor suppressive pathways.

Referring to study limitations and future directions that warrant consideration, the relatively small sample size and lack of an independent validation cohort may limit the generalizability of our findings and introduce potential biases related to age and comorbidities between HCC cases and controls. Additionally, the use of healthy controls rather than patients with end-stage liver disease (ESLD) may confound the specificity of the observed circRNA alterations. Given that most HCC cases arise from a background of cirrhosis or chronic liver disease, future studies should include ESLD-matched controls for more accurate biomarker validation.

Another limitation is the exclusive reliance on plasma samples, which—while offering a non-invasive diagnostic approach—precludes direct comparison with tumor tissue and may limit insights into tumor-specific expression profiles. Furthermore, functional validation of the proposed circ_0027478/miR-1236-3p/AFP regulatory axis is lacking. Experimental approaches such as dual-luciferase assays, RNA pull-down, and loss/gain-of-function studies are needed to confirm molecular interactions and clarify downstream effects on key signaling pathways like PI3K/Akt in HCC progression.

Lastly, although this study focuses on the predictive value of candidate biomarkers, it does not address the interpretability of AI-based biomedical models. Improving model transparency is critical for clinical translation. Future work should consider incorporating interpretable AI tools such as knowledge graphs and graph neural network (GNN) visualizations to elucidate biological relationships and enhance the reliability and usability of predictive models in HCC and beyond.

## 4. Materials and Methods

### 4.1. Patients

This research involved 150 Egyptian individuals, consisting of 100 cases of HCC (58 males and 42 females), aged between 34 to 89 years, with a mean age of 59.2 ± 0.8 years. Additionally, there were 50 healthy controls matched to the patients in terms of age and sex (27 males and 23 females), with ages falling between 45 to 65 years and a mean age of 60.7 ± 0.96 years. None of them had a history of chronic illnesses including any chronic liver diseases, all had just received new diagnoses, all had perfectly normal kidney function tests, and none had received radiation or chemotherapy.

Patients diagnosed with HCC were selected from the inpatient and outpatient clinics of the National Cancer Institute (NCI) in Cairo, Egypt, between March 2022 and January 2023. A computed tomography scan was advised for screening the individuals who were believed to have HCC since it may reveal the existence of this condition. In addition, concerns about symptoms and indicators of HCC, such as significant unexplained weight loss, enlargement of the liver, and gastrointestinal hemorrhage, were discussed. The pathologist confirmed positive findings.

The guidelines that the research relies on are the EASL guidelines that rely heavily on imaging, especially for patients at risk of HCC, such as those with cirrhosis. They allow for the diagnosis of HCC using non-invasive imaging, with criteria including the use of CT or MRI to identify lesions exhibiting arterial enhancement and venous washout. While AFP levels may be utilized in conjunction with imaging for screening or diagnosis, they are not definitive on their own and should be interpreted alongside other clinical and imaging findings [76].

A comprehensive compilation of clinical data was conducted for each patient, encompassing their complete medical history, complete blood count (CBC), liver biochemical profile, kidney function tests, coagulation profile, and ALP activity, as well as the clinical characteristics of patients with HCC, such as tumor size, lymph node status, tumor spreading, and grade. These data were obtained from the medical records. Furthermore, the classification of patients as metastatic or non-metastatic was contingent upon the use of computerized tomography scans, positron emission scanning, or imaging with magnetic resonance. To gather information on epidemiology, we conducted qualitative exploratory interviews with individuals who were relatives of the first degree. This method was chosen to provide a more comprehensive understanding of the patients’ backgrounds, including variables such as age, gender, smoking habits, and cancer history, which are not always captured in medical records.

The control group, a crucial part of our study, comprised healthy individuals attending the hospital for routine check-ups or non-cancer-related medical issues. These individuals were matched with the HCC patient cohort according to age and gender, ensuring comparability. The controls were selected according to specific inclusion and exclusion criteria to confirm the absence of any history of liver disease, cancer, or other significant medical conditions that might confound the study results. Their role in our research cannot be overstated.

AST, ALT, HCV antibodies, INR, PT, albumin, and bilirubin were collected from patients’ routine medical records. These data, which were part of the standard clinical assessments conducted by the hospital’s laboratory team, were easily accessible from the hospital’s electronic database, ensuring the transparency of our data collection process.

### 4.2. Blood Specimens

Blood samples (5 mL) were obtained from all participants in the morning (between 8:00 and 10:00 a.m.) to reduce circadian variations in RNA expression at the National Cancer Institute (NCI) in Cairo, Egypt, under the supervision of the Department of Clinical Pathology. The samples were collected in ethylenediaminetetraacetic acid (EDTA) vacutainers and incubated for 30 min at room temperature before centrifugation at 2000 g (≈4000 rpm) for 10 min to separate plasma from whole blood. The plasma was subsequently divided into two aliquots: one for RNA extraction and the other for AFP concentration assessment. Both aliquots were promptly stored at −80 °C to maintain their integrity [77]. This research and all empirical investigations were conducted in accordance with relevant legislation and ethical guidelines. Written informed consent was obtained from all participants, including patients, controls, and their legal representatives. The study was approved by the Medical Ethical Committee at the Faculty of Pharmacy (Girls), Al-Azhar University (Approval No: REC: 327:11\2021). All procedures adhered to the ethical standards outlined in the Declaration of Helsinki.

### 4.3. Selection of circRNAs and Prediction of circRNA–miRNA Interactions

These circRNAs were identified through data analysis reports (GSE94508 and GSE97332) retrieved from the database of the Gene Expression Omnibus (GEO) [42], where both were consistently dysregulated in HCC tissue samples [42]. These datasets were rigorously analyzed using GEO2R, with stringent criteria (*p*-value < 0.05 and |log fold change| > 1.0) to ensure statistical significance. This thorough process underscores the scientific validity of our research. Furthermore, these circular RNAs were consistently dysregulated across both datasets, suggesting their potential role in HCC progression [42].

To further explore the biological significance of these circRNAs, we performed bioinformatic analyses to predict their interactions with candidate microRNAs (miRNAs). The web resources circinteractome (https://circinteractome.nia.nih.gov/ (accessed on 5 May 2021)) and miRDB (http://www.mirdb.org/ (accessed on 5 May 2021)) were used [78,79,80]. miRDB and TargetScan were used to predict the selected miRNA target genes (http://www.targetscan.org/mamm_31/ (accessed on 6 May 2021)). In contrast, mirTarBase (http://mirtarbase.cuhk.edu.cn/ (accessed on 6 May 2021)) was used to obtain empirically firmly supported target genes (via Western blot, qPCR reporter assays) of the selected miRNA [81,82,83].

This prediction aligns with previous evidence highlighting miR-1236-3p as a tumor suppressor involved in regulating the PI3K/Akt pathway via targeting AFP. Therefore, circ_0027478/miR-1236-3p interaction was prioritized for further investigation, as it suggests a potential competitive endogenous RNA (ceRNA) mechanism through which circ_0027478 may contribute to HCC pathogenesis. This integrated rationale and prediction strategy form the basis for our focus on these specific ncRNA markers in the current study.

### 4.4. Sample Size Calculation

To calculate the sample size for testing circRNA in plasma, we utilized the statistical software G*Power, version 3.1.9.7, accessed on 1 November 2021, considering several key factors. These included the effect size, which reflects the difference of mean in circRNA expression levels between the patient and control groups, as well as the type I error rate, set at 0.05, indicating a 5% likelihood of incorrectly detecting a difference. The type II error (β) was set at 0.2, corresponding to a power of 80%, ensuring an 80% probability of detecting a true effect if one exists. These parameters, along with the standard deviation of circRNA levels in plasma obtained from previous studies, were used to accurately determine the sample size required to identify significant differences in circRNA expression [72,84].

The analysis indicated that each group should contain at least 51 participants, with a total minimum sample size of 102 participants to achieve sufficient statistical power. However, to enhance the robustness of our findings and account for potential variability, we included 150 participants (100 HCC patients and 50 controls) in our study. This ensures greater reliability and generalizability of the results.

### 4.5. Assessment of Plasma circ_0009910, circ_0027478, and miR-1236 Using RT-qPCR

To extract total RNA, 100 μL of plasma was treated with QIAzol lysis reagent in QIAGEN’s miRNeasy extraction kit, cat no. 217004 (QIAGEN, Valencia, CA, USA). The POLARstar^®^Omega Nanodrop (BMG Labtech, Ortenberg, Germany) was used to assess RNA purity and concentration, ensuring that only high-quality RNA samples with an A260/A280 ratio of at least 1.8 were used for cDNA synthesis of the chosen circular RNAs and miRNAs.

To create cDNA of hsa_circ_0009910 and hsa_circ_0027478, 10 μL of total RNA (0.01 pg–0.5 μg) was used, following the rules of RevertAid First Strand cDNA Synthesis, cat no. K1622 (Thermo Fisher Scientific, Waltham, MA, USA). Thermal cycler settings were used for reverse transcription (5 min at 25 °C for primer annealing, 60 min at 42 °C for processing, and 5 min at 70 °C for termination).

Using glyceraldehyde 3-phosphate dehydrogenase (GAPDH) as an internal control for normalization, the expression profile of both circular RNAs in the plasma of patients and controls was investigated [85]. The specific custom-made divergent primers (Sangon Biotech, Shanghai, China) predesigned for the circular RNAs were employed utilizing the software of primer3 web (https://primer3.ut.ee/ (accessed on 1 June 2021)). Before modification, the tool NCBI primer-blast (https://www.ncbi.nlm.nih.gov/tools/primer-blast/ (accessed on 1 June 2021)) was utilized to verify the specificity of the divergent primers, with an access date of 5 June 2023. Table 5 mentions primer sequences. In summary, qPCR was carried out with a Bio-Rad miniopticon instrument equipped with software version 3.1. The experiment included the preparation of 20 μL mixtures of reactions employing the PCR SensiFAST^TM^ SYBR No-ROX Kit, cat no. BIO-98005 (Bioline [Life Science ^®^ Company], Seoul, Republic of Korea), along with reverse and forward primers as prescribed beforehand. The thermocycler operated for two minutes at a temperature of 95 °C; next came 40 cycles consisting of five seconds at 95 °C, ten at 60 °C, and ten at 72 °C.

The miScript II RT Kit, a reliable product from QIAGEN, cat no. 218161, was used for the transcription of miRNAs. The procedure, following the manufacturer’s instructions, included the use of total RNA (0.1 μg) in 20 μL (L) assays. The cDNA specimens were diluted and generated using the miScript SYBR Green PCR set, (cat no. 218073, QIAGEN, Hilden, Germany) and the provided miScript Global Primer (reverse primer). Additionally, miScript Primer (forward primer) tests for hsa-miR-1236-3p and the housekeeping miScript PCR reference miRNA SNORD68 (QIAGEN, Hilden, Germany) were utilized [86].

Multiple studies have provided empirical evidence supporting the efficacy of SNORD68 as an internal reference for the normalization of miRNAs. Multiple works of research have shown empirical support for the use of this method as a benchmark for the comparative measurement of miRNAs since it demonstrates consistent and comparable levels of expression in the plasma of individuals with illnesses and control groups [77,87]. In summary, the Bio-Rad miniopticon instrument equipped with software version 3.1. was utilized for conducting real-time PCR in 20 μL reaction mixtures that were produced following earlier descriptions [88,89]. For every PCR run, the following thermal conditions were used: 95 °C for thirty seconds, 40 cycles of 94 °C for fifteen seconds, 55 °C for 30 s, and 70 °C for 30 s.

Analysis of the melting curve was used to verify the specificity of the PCR outcomes. It was estimated what the gene expression was in relation to internal control (2^−ΔΔCt^). The fold change was expressed by applying the 2^−ΔΔCt^ technique for relative quantification [90].

### 4.6. Assessment of Serum AFP, Using ELISA

The AFP levels were measured using an ELISA kit (AFP ELISA Kit, Creative Diagnos-tics, cat no. DEIA080, New York, NY, USA). The minimum detection level for AFP was 5 ng/mL, and the inter-assay coefficient of variation was <10%, while the intra-assay coefficient of variation was <5%. These values ensure the assay’s reliability and consistency.

For samples with AFP concentrations greater than 400 ng/mL, dilution is recommended, and the result is adjusted by multiplying the value by the dilution factor [91].

### 4.7. The Difference Between Staging Systems in the Current Study

To ensure a comprehensive assessment of HCC, multiple staging systems were utilized. The TNM system classifies tumors based on size, nodal involvement, and distant metastasis, providing an anatomical overview of disease progression. The BCLC staging system incorporates tumor burden, liver function (Child–Pugh score), and patient performance status, offering guidance on treatment allocation. The Child–Pugh classification evaluates liver function based on bilirubin, albumin, INR, ascites, and hepatic encephalopathy, and it is widely used to assess the severity of liver disease. Lastly, the Okuda staging system considers tumor size, ascites, albumin, and bilirubin levels, making it particularly relevant in advanced liver disease. These complementary systems allow for a multidimensional evaluation of HCC prognosis and treatment planning [92,93,94,95].

### 4.8. Statistical Methods

Data were analyzed to analysis using the Statistical Package for the Social Sciences (SPSS.22) developed by IBM in the United States, as well as the Prism 9.5.1 statistics tool developed in the USA. In this study, categorical data were represented using numbers and percentages, while numerical data were analyzed using statistical measures such as mean ± standard error, median (with 25–75% percentiles), or range, if appropriate. To assess the normality of the data, D’Agostino–Pearson, Kolmogorov–Smirnov, and Shapiro–Wilk analyses were performed. The statistical methods used to compare numerical variables, where appropriate, included the Mann–Whitney U test, *t*-test, Kruskal–Wallis test followed by Dunn’s test, one-way ANOVA followed by Tukey’s test, or post hoc test. Fisher’s exact analysis was performed to compare categorical data within the study. Receiver operating characteristic (ROC) analysis was used to assess the diagnostic performance of the markers under investigation, and subsequently, the area under the curve (AUC) was calculated. The AUCs were categorized into three groups, AUC = 0.6 to 0.7, 0.7 to 0.9, and >0.9, to specify the biomarker as a significant, promising, and outstanding discriminator, respectively. This study used univariate and multivariate logistic regression analysis to classify the predictor markers linked to HCC versus the control group. A stepwise-forward multivariate analysis was conducted, using pertinent significant factors obtained from the univariate analysis, to obtain the final variables linked to the probability of receiving a diagnosis of HCC. Spearman’s rho correlation coefficient was used to assess the correlations among the measurements. Statistical significance was established for the findings when the two-tailed *p*-value of the test was found to be below the threshold of 0.05.

## 5. Conclusions

This study highlights the diagnostic and prognostic potential of plasma circ_0009910, circ_0027478, and miR-1236-3p as promising non-invasive biomarkers for HCC. Among these, circ_0009910 demonstrated the highest diagnostic accuracy, significantly outperforming AFP and showing robust associations with tumor size, stage, metastasis, and cirrhosis severity. The integration of these ncRNAs with AFP further enhanced diagnostic performance, underscoring the value of multi-biomarker panels in improving early detection strategies.

From a prognostic standpoint, circ_0009910 also distinguished between metastatic and non-metastatic cases with notable specificity and sensitivity, suggesting its potential utility in patient risk stratification and disease monitoring. The observed correlations between circRNAs and key clinical features reinforce their relevance in the molecular pathology of HCC.

However, while these findings are encouraging, they must be interpreted with caution. The lack of functional validation and the limited sample size call for further investigation. Future research should include mechanistic studies—such as dual-luciferase assays and pathway analyses—to validate the proposed circ_0027478/miR-1236-3p/AFP axis and its regulatory impact on tumor progression via pathways like PI3K/Akt. Moreover, large-scale, multi-center cohorts that include ESLD-matched controls are essential to confirm the clinical applicability of these biomarkers across diverse patient populations.

In parallel, enhancing the interpretability of predictive models through tools such as knowledge graphs and GNN visualization will be crucial to ensure the translational success of such biomarker-driven approaches. Collectively, our findings lay a foundational framework for the development of more precise, minimally invasive diagnostic and prognostic tools in HCC, opening avenues for personalized medicine and improved patient outcomes.

## Figures and Tables

**Figure 1 ijms-26-04842-f001:**
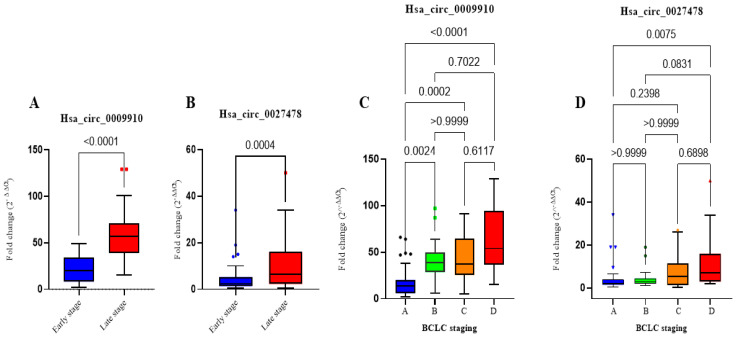
Plasma levels of hsa_circ_0009910 and hsa_circ_0027478 in the studied groups. (**A**) Fold change of plasma circ_0009910 expression levels in patients with early stage of HCC (*n* = 53) using TNM staging compared with late stage of HCC (*n* = 47). (**B**) Fold change of plasma circ_0027478 expression levels in patients with early stage of HCC vs. late stage of HCC. (**C**) Fold change of plasma expression levels of hsa_circ_0009910 among four subgroups of HCC in relation to the BCLC staging system. (**D**) Fold change of plasma expression levels of hsa_circ_0027478 among four subgroups of HCC in relation to the BCLC staging system. The data are presented as a box blot, with the box representing the 25–75% percentiles, the line inside the box indicating the median, and the top and lower lines representing 10–90%. A *p*-value of < 0.05 indicates statistical significance. BCLC staging system: Barcelona Clinic Liver Cancer, HCC: hepatocellular carcinoma, TNM: tumor, node, metastasis staging. *p*-values < 0.05 are considered significant.

**Figure 2 ijms-26-04842-f002:**
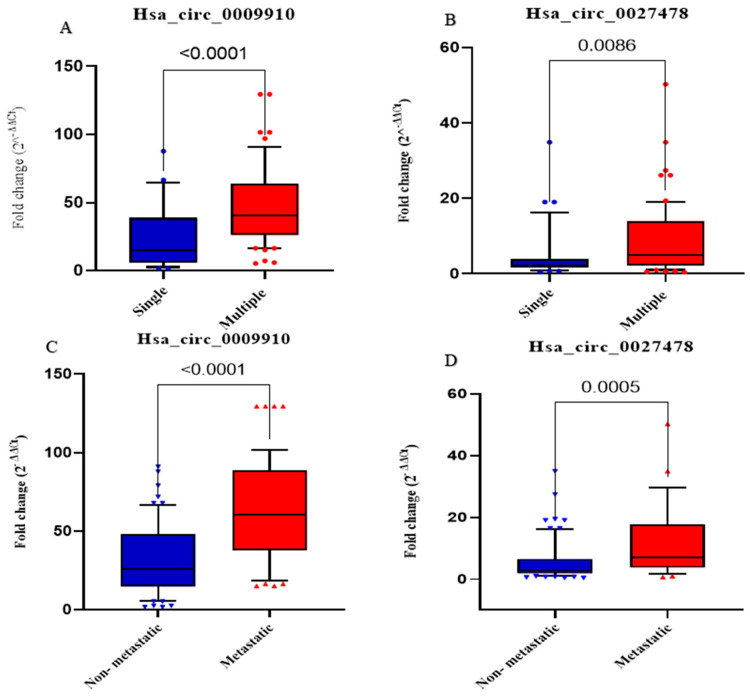
The plasma concentrations of circ_0009910 and circ_0027478 were measured in the populations under investigation. (**A**) The statistically significant difference in the fold change of plasma circ_0009910 expression levels across patients with varying quantities of tumor lesions. These patients were divided into two groups: single (*n* = 32) and multiple (*n* = 68). (**B**) The differential expression levels of hsa_circ_0027478 demonstrated the ability to distinguish between the two subgroups according to the number of tumor lesions. (**C**) Fold change of circ_0009910 expression levels in non-metastatic HCC (*n* = 75) versus metastatic HCC (*n* = 25). (**D**) Fold change of circ_0027478 expression levels in non-metastatic HCC vs. metastatic HCC. *p*-values < 0.05 are considered significant.

**Figure 3 ijms-26-04842-f003:**
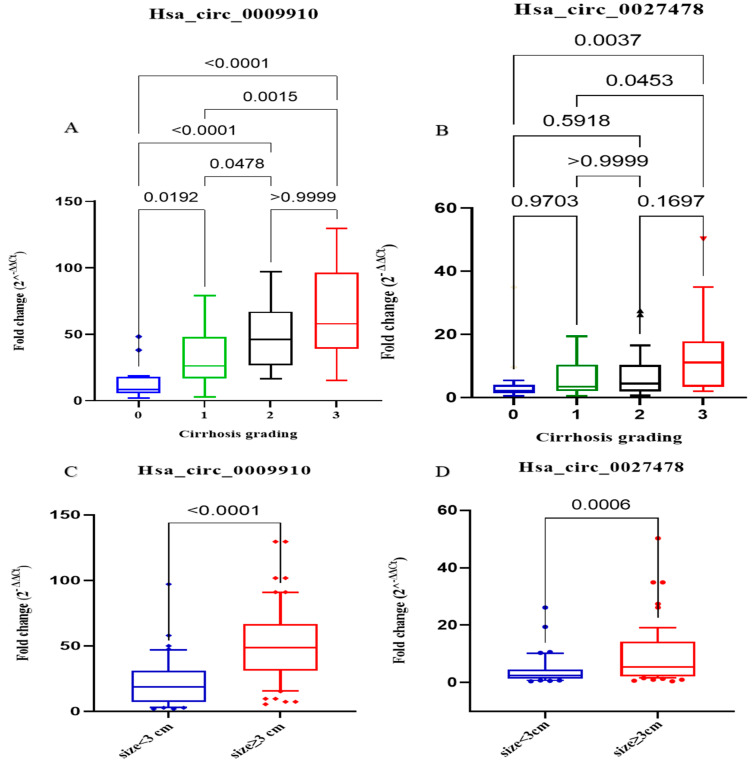
Plasma concentrations of hsa_circ_0009910 and hsa_circ_0027478 were evaluated in subgroups depending on cirrhosis and target tumor size. (**A**) The fold change in plasma hsa_circ_0009910 expression levels may be used to distinguish between all stages of cirrhosis, except stages 2 and 3. In these stages, patients with HCC were grouped into 4 groups based on cirrhosis staging. {0: absence of fibrosis (*n* = 16); grade 1: the presence of mild fibrosis (*n* = 39); grade 2: the presence of moderate fibrosis (*n* = 28); Grade 3: the presence of severe fibrosis or cirrhosis (*n* = 17)}. (**B**) The fold change in expression levels of hsa_circ_0027478 may effectively distinguish between stage 0 and stage 3 and between stage 1 and stage 3. (**C**,**D**) The statistical significance of the difference in fold change of plasma hsa_circ_0009910 and hsa_circ_0027478 expression levels across patients with different tumor sizes. The patients were categorized into two groups based on their size: those with a size less than 3 (*n* = 40) and those with a size more than or equal to 3 (*n* = 60). *p*-values < 0.05 are considered significant.

**Figure 4 ijms-26-04842-f004:**
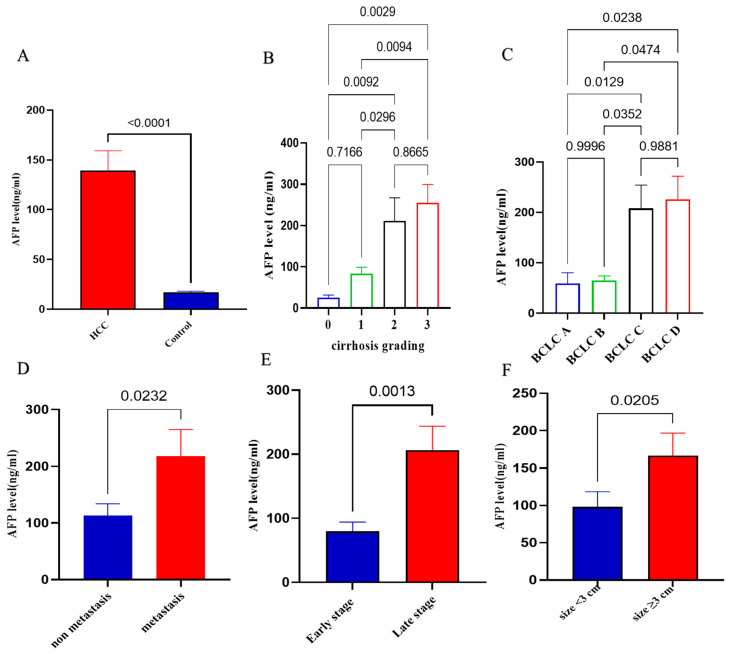
AFP plasma levels in the groups under investigation. (**A**) A statistical significance of *p* < 0.0001 is shown by data provided as mean ± SE; HCC (*n* = 100) and 50 matched controls. (**B**) AFP levels may effectively distinguish different stages of cirrhosis. (**C**) AFP levels among four subgroups of HCC in relation to the BCLC staging system; BCLC staging system: Barcelona Clinic Liver Cancer; early stage (Stage A and Stage B), late stage (Stage C and Stage D). (**D**) AFP levels in non-metastatic HCC (*n* = 75) versus metastatic HCC (*n* = 25) (*p* = 0.023). (**E**) AFP levels in patients with early stage of HCC (*n* = 49) compared with late stage of HCC (*n* = 51) according to TNM staging system (*p* = 0.0013). (**F**) The statistical significance difference in plasma AFP levels across patients with different tumor sizes (*p* = 0.0205).

**Figure 5 ijms-26-04842-f005:**
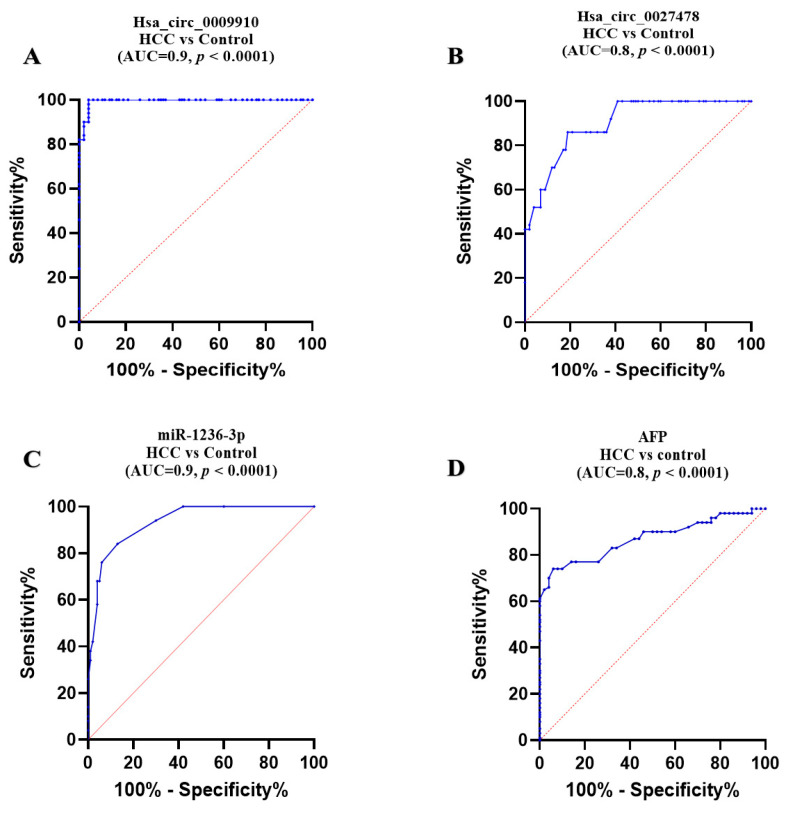
Diagnostic accuracy of the studied parameters. (**A**) circ_0009910, (**B**) circ_0027478, (**C**) miRNA-1236, (**D**) AFP: alpha-fetoprotein. Using ROC curve analysis. HCC: hepatocellular carcinoma (*n* = 100); healthy controls (*n* = 50). AUC: area under the curve. *p*-values < 0.05 are considered significant.

**Figure 6 ijms-26-04842-f006:**
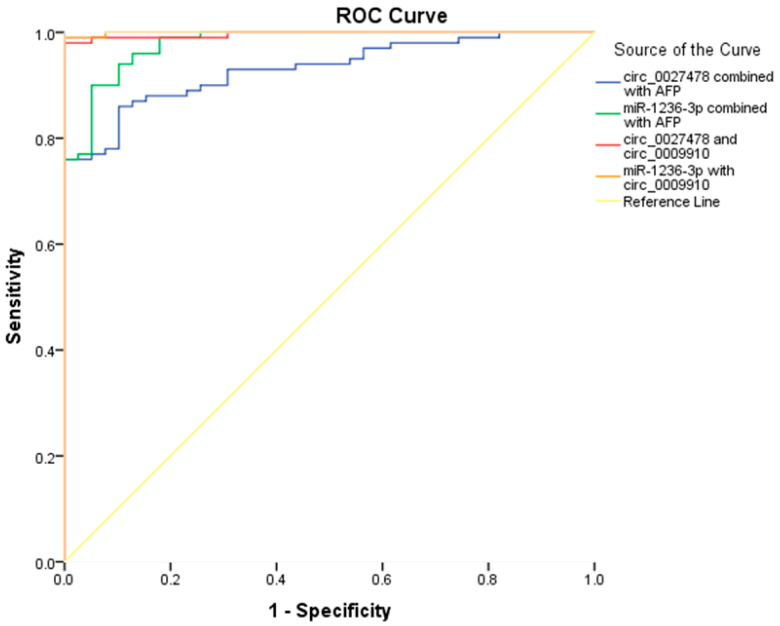
Combined ROC curves illustrating the diagnostic performance of various biomarker combinations for HCC. Model 1: AFP + circ_0027478; Model 2: AFP + miR-1236-3p; Model 3: circ_0027478 + circ_0009910; Model 4: miR-1236-3p + circ_0009910. The integration of circular RNAs and microRNAs significantly enhanced the diagnostic power of AFP in distinguishing HCC cases from controls.

**Figure 7 ijms-26-04842-f007:**
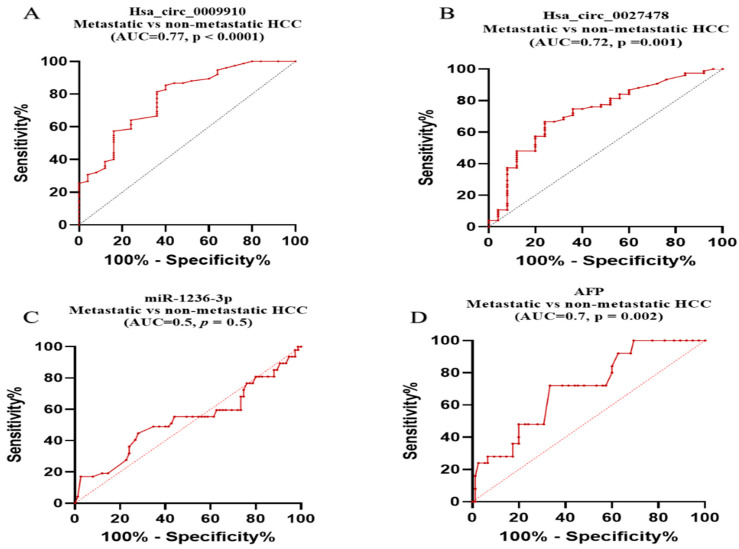
The prognostic power of the studied parameters. (**A**) circ_0009910, (**B**) circ_0027478, (**C**) miRNA-1236, (**D**) AFP: alpha-fetoprotein. Using ROC curve analysis, metastasis (*n* = 25), non-metastasis (*n* = 75). AUC: area under the curve. *p*-values < 0.05 are considered significant.

**Figure 8 ijms-26-04842-f008:**
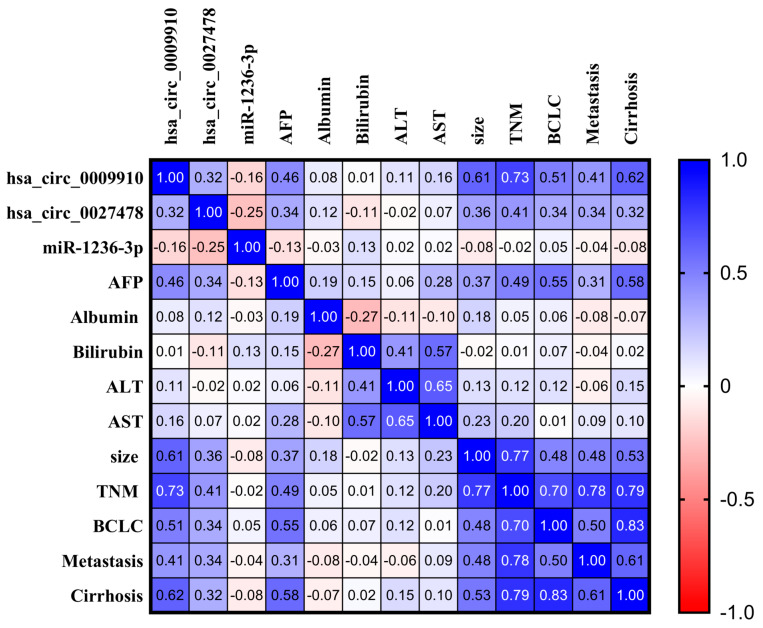
Correlation analysis among plasma markers and their relationship with clinical and laboratory data among patients with HCC. A correlation map based on a blue–red (cold–hot) continuum. The blue tint suggests a correlation around 1, whereas the color red implies a relationship near −1. White denotes a correlation approaching zero. Associations are calculated using the Spearman regression coefficient. AFP: alpha-feto protein; ALT: alanine transaminase; AST: aspartate transaminase; BCLC: staging system—Barcelona Clinic Liver Cancer; TNM: tumor, node, metastasis staging.

**Table 1 ijms-26-04842-t001:** Study groups’ demographic, laboratory, and clinicopathological data.

			Healthy Controls (*n* = 50)	HCC (*n* = 100)	*p*-Value
Age in years Age range			55.4 ± 0.75 (45–65)	59.2 ± 0.8 (34–89)	0.22
**Sex, *n* (%)**	Male		27 (54%)	58 (58%)	0.64
	Female		23 (46%)	42 (42%)	
ALT (IU/L)			16.6 ± 0.8	62.3 ± 5.8	<0.001
AST (IU/L)			18 ± 0.6	102.2 ± 12.1	<0.001
AFP (ng/mL)			3.7 ± 0.2	139.2 ± 20	<0.001
Albumin (gm/dL)			4.6 ± 0.12	2.6 ± 0.06	<0.001
Bilirubin (mg/dL)			0.5 ± 0.03	6.8 ± 1.4	<0.001
PT (seconds)			12.1 ± 0.1	18.2 ± 0.91	<0.001
INR			0.9 ± 0.01	1.5 ± 0.07	<0.001
**HCV Ab**, ***n* (%)**		Positive	-	79, (79%)	
		Negative		21, (21%)	
**Tumor anatomical site, *n* (%)**					
Left lobe (LT)			-	44, (44%)	-
Right lobe (RL)			-	35, (35%)	-
LT&RL			-	21, (21%)	-
**Child staging, *n* (%)**					
A			-	19, (19%)	-
B			-	27, (27%)	-
C			-	54, (54%)	-
**TNM staging system,** ***n* (%)**					
Stage I, II (Early)			-	53, (53%)	-
Stage III, IV (Late)			-	47, (47%)	-
**BCLC staging system** **, *n* (%)**					
A			**-**	28, (28%)	-
B			**-**	21, (21%)	-
C			**-**	34, (34%)	-
D			**-**	17, (17%)	-
**Okuda staging,** ***n* (%)**					
1			**-**	18, (18%)	-
2			**-**	34, (34%)	-
3			**-**	48, (48%)	-
**Number of lesion** **, *n* (%)**					
Single			**-**	32, (32%)	-
Multiple			**-**	68, (68%)	-
**Lymph node metastasis,** ***n* (%)**					
Present			-	42, (42%)	-
Absent			-	58, (58%)	-
**Distant metastasis** **, *n* (%)**					
Present			-	25, (25%)	-
Absent			-	75, (75%)	-
**Cirrhosis** **, *n* (%)**					
0			-	16, (16%)	-
1			-	39, (39%)	-
2			-	28, (28%)	-
3			-	17, (17%)	-
**Tumor size**,***n* (%)**					
<3			-	40, (40%)	
≥3			-	60, (60%)	

Data are provided as mean ± SE or number and percentage (%). Independent t-tests were used to conduct the statistical analysis. Categorical data were compared using the chi-square test. Cirrhosis is classified into four stages (0: no fibrosis; grade 1: mild fibrosis; grade 2: moderate fibrosis; grade 3: severe fibrosis or cirrhosis), and the BCLC staging system: Stage A (early stage), Stage B (intermediate stage), Stage C (advanced stage), and Stage D (severe liver damage [34]. The TNM (tumor, node, metastasis) staging system was used to stage HCC. AFP, alpha-fetoprotein; ALT, alanine transaminase; AST, aspartate transaminase; BCLC, Barcelona Clinic Liver Cancer; HCV, hepatitis C virus; INR, international normalized ratio; PT, prothrombin time; TNM, tumor, node, metastasis staging system. *p*-values < 0.05 are considered significant.

**Table 2 ijms-26-04842-t002:** The fold change in plasma expression levels of hsa_circ_0009910, hsa_circ_0027478, and hsa_miR_1236 in malignancy patients and controls.

	HCC (*n* = 100)	Control (*n* = 50)	*p*-Value
hsa_circ_0009910	35.6 (16.7–54.2)	0.57 (0.18–1.2	<0.001
hsa_circ_0027478	3.8 (1.9–10.5)	0.66 (0.1–1.3)	<0.001
hsa_miR_1236-3p	0.13 (0.05–0.3)	0.8 (0.5–1.2)	<0.001

The data are shown using the median and interquartile range (IQR). Differential fold change compared to the healthy group. To compute the 2^^−∆∆Ct^ for control samples, each Ct value was subtracted from the average Ct of the control group. The *p*-values shown in bold indicate significant difference, namely *p* < 0.05.

**Table 3 ijms-26-04842-t003:** Logistic regression assessment to anticipate the risk of HCC.

Parameter	Coefficient	SE	*p*-Value	OR	OR (95% CI)
**Univariate**					
Hsa_circ_0009910	1.185	0.3	**0.0001**	3.27	1.804–5.936
Hsa_circ_0027478	1.3530	0.262	**<0.0001**	3.86	2.311–6.475
miRNA-1236	−6.4	1.05	**<0.0001**	0.001	0.0002–0.012
AFP	0.0992	0.02	**<0.0001**	1.10	1.052–1.158
**Multivariate**					
Hsa_circ_0009910	1.072	0.66	0.109	2.93	0.786–10.86
Hsa_circ_0027478	0.928	1.04	0.374	2.52	0.325–19.65
miRNA-1236	−14.682	11.033	0.183	0.00	0.001–1035.8152
AFP	0.0703	0.203	0.729	1.07	0.719–1.59
Constant	−1.08				

HCC: hepatocellular carcinoma (*n* = 100); control (*n* = 50). AFP: alpha-fetoprotein; CI: confidence interval; OR: odds ratio; SE: standard error. *p* values in bold are statistically significant (*p* < 0.05).

**Table 4 ijms-26-04842-t004:** Correlation between the studied plasma markers and the clinical and laboratory data among patients with HCC.

		circ_0009910	circ_0027478	miR-1236-3p	AFP	Albumin	Bilirubin	ALT	AST	Size	TNM	BCLC	Metastasis	Cirrhosis
**circ_0009910** **(fold change)**	**r**	1.000	0.325	−0.165	0.455	0.076	0.010	0.112	0.164	0.608	0.728	0.507	0.415	0.616
	** *p* **		**0.001**	**0.101**	**<0.001**	0.450	0.925	0.276	0.103	**<0.001**	**<0.001**	**<0.001**	**<0.001**	**<0.001**
**circ_0027478** **(fold change)**	**r**	0.325	1.000	−0.252	0.341	0.119	−0.105	−0.020	0.065	0.363	0.412	0.338	0.342	0.322
	** *p* **	**0.001**		**0.011**	**0.001**	0.237	0.289	0.843	0.518	**<0.001**	**<0.001**	**<0.001**	**<0.001**	**<0.001**
**miR-1236-3p** **(fold change)**	**r**	−0.165	−0.252	1.000	−0.132	−0.031	0.130	0.015	0.018	−0.083	−0.015	0.047	−0.044	−0.083
	** *p* **	0.101	**0.011**		0.189	0.762	0.197	0.880	0.859	0.414	0.881	0.644	0.664	0.413
**AFP(ng/mL)**	**r**	0.455	0.341	−0.132	1.000	0.190	0.152	0.060	0.283	0.366	0.494	0.544	0.306	581
	** *p* **	**<0.001**	**0.001**	0.189		0.058	0.131	0.553	**0.004**	**<0.001**	**<0.001**	**<0.001**	**0.002**	**<0.001**
**Albumin (g/dl)**	**r**	0.076	0.119	−0.031	0.190	1.000	−0.272	−0.110	−0.097	0.176	0.049	0.065	−0.084	−0.070
	** *p* **	0.450	0.237	0.762	0.058		**0.006**	0.275	0.339	0.081	0.629	0.523	0.408	0.491
**Bilirubin (mg/dl)**	**r**	0.010	−0.105	0.130	0.152	−0.272	1.000	0.408	0.572	−0.022	0.013	0.067	−0.037	0.024
	** *p* **	0.925	0.298	0.197	0.131	**0.006**		**<0.001**	**<0.001**	0.831	0.899	0.511	0.298	0.814
**ALT(U/L)**	**r**	0.112	−0.020	0.015	0.060	−0.110	0.408	1.000	0.645	0.132	0.122	0.124	−0.058	0.146
	** *p* **	0.267	0.843	0.880	0.553	0.275	**<0.001**		**<0.001**	0.189	0.228	0.220	0.566	0.148
**AST(U/L)**	**r**	0.164	0.065	0.018	0.283	−0.097	0.572	0.645	1.000	0.233	0.205	0.012	0.092	0.103
	** *p* **	0.103	0.518	0.859	0.004	0.339	**<0.001**	**<0.001**		**0.020**	**0.041**	0.909	0.360	0.308
**Size(cm)**	**r**	0.608	0.363	−0.083	0.366	0.176	−0.022	0.132	0.233	1.000	0.765	0.482	0.484	0.532
	** *p* **	**<0.001**	**<0.001**	0.414	**<0.001**	0.081	0.831	0.189	**0.020**		**<0.001**	**<0.001**	**<0.001**	**<0.001**
**TNM**	**r**	0.728	0.412	−0.015	0.494	0.049	0.013	0.122	0.205	0.765	1.000	0.698	0.778	0.793
	** *p* **	**<0.001**	**<0.001**	0.881	**<0.001**	0.629	0.899	0.228	**0.041**	**<0.001**		**<0.001**	**<0.001**	**<0.001**
**BCLC**	**r**	0.507	0.338	0.047	0.554	0.065	0.067	0.124	0.012	0.482	0.698	1.000	0.496	0.833
	** *p* **	**<0.001**	**0.001**	0.644	**<0.001**	0.523	0.511	0.220	0.909	**<0.001**	**<0.001**		**<0.001**	**<0.001**
**Metastasis**	** *r* **	0.415	0.342	−0.044	0.306	−0.084	−0.037	−0.058	0.092	0.484	0.778	0.496	1.000	0.606
	** *p* **	**<0.001**	**0.001**	0.664	**0.002**	0.408	0.716	0.566	0.360	**<0.001**	**<0.001**	**<0.001**	**0.001**	**<0.001**
**Cirrhosis**	** *r* **	0.616	0.322	−0.083	0.581	−0.070	0.024	0.146	0.103	0.532	0.793	0.833	0.606	1.000
	** *p* **	**<0.001**	**0.001**	0.413	**<0.001**	0.491	0.814	0.148	0.308	**<0.001**	**<0.001**	**<0.001**	**<0.001**	

Spearman correlation was used to measure correlation. r: Spearman’s rho coefficient. *p* values in bold indicate statistical significance (*p* < 0.05). AFP, alpha-fetoprotein; ALT, alanine transaminase; AST, aspartate transaminase; BCLC, Barcelona Clinic Liver Cancer; HCV, hepatitis C virus; TNM, tumor, node, metastasis staging system.

**Table 5 ijms-26-04842-t005:** Primer sequences that were customized for the investigation. GAPDH: glyceraldehyde 3-phosphate dehydrogenase.

Gene	Primer Sequence	
has_circ_0009910	F.	5′-GCAGAACTGGACCCCGTTACC-3′
	R.	5′-CAGGGACATTGCGCGGCCAA-3′
Hsa_circ_0027478	F.	5′-CCATTGCCTGGAGTTGGCT-3′
	R.	5′-CCACAGCGTTTACAGAGTCG-3′
GAPDH	F.	5′-CCCTTCATTGACCTCAACTA-3′
	R.	5′-TGGAAGATGGTGATGGGATT-3′

F. indicates forward primer, and R. refers to reverse primer.

## Data Availability

All the data and material used are available in the manuscript.

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
