# Peer review of "Preliminary Evaluation of Plasma circ_0009910, circ_0027478, and miR-1236-3p as Diagnostic and Prognostic Biomarkers in Hepatocellular Carcinoma"

_ijms, 2025, doi:10.3390/ijms26104842_

Round 1

Reviewer 1 Report (New Reviewer)

Comments and Suggestions for Authors

1. AFP, as a traditional biomarker, was not discussed in the article for its clinical significance in combination diagnosis with circRNAs/miR-1236-3p

2. The article only predicts the interaction between circ_0027478 and miR-1236-3p through bioinformatics, lacking functional experimental verification (such as dual luciferase reporter genes, RNA pull-down, etc.). Furthermore, it has not been elucidated how circRNAs affect HCC progression by regulating downstream signaling pathways such as PI3K/Akt.

3. The healthy control group (50 cases) did not include patients with cirrhosis or chronic liver disease, while 79% of HCC patients were HCV positive, which may lead to an overestimation of the specificity of the markers. In actual clinical scenarios, HCC often occurs on the basis of cirrhosis, and the lack of cirrhosis control will limit the clinical applicability of the results.

4. The RNA extraction and qPCR steps did not mention how to avoid confusion between circRNA and linear RNA (such as RNase R treatment), nor did they specify whether primer design crosses the reverse splicing site to specifically amplify circRNA.

This study reports for the first time the diagnostic and prognostic potential of plasma circ_0009910 and circ_0027478 in HCC. The data is innovative, but further refinement is needed in terms of mechanism validation, sample representativeness, and clinical practicality.

Author Response

Reviewer 2 Report (New Reviewer)

Comments and Suggestions for Authors
  1. Introduction

This section is overly lengthy, with redundant background information and an unfocused presentation of the core scientific question. It is recommended to simplify the expression, for example, by appropriately reducing the background discussion on other circRNAs (such as circMDK and circ_0000098) to avoid diverting from the main research focus. Moreover, this section lacks a summary of previous findings on circ_0009910 and circ_0027478 in hepatocellular carcinoma studies. In addition, the final paragraph should highlight the innovation of the present study to enhance its appeal.

  1. Results

The description in this section is somewhat disjointed, and the grouping and comparisons are not systematic, which may cause confusion for readers. It is recommended to consolidate all expression difference analyses together, and group all correlation analyses between expression levels and disease characteristics separately. In addition, while the manuscript mentions circ_0009910, circ_0027478, and miR-1236-3p, one of these is missing in some of the correlation analyses between expression levels and disease characteristics. Moreover, although different tumor staging systems are mentioned, the manuscript does not explain the differences among these staging methods.

  1. Discussion

This section lacks a discussion on AI interpretability, which remains a challenge for food prediction models. It is recommended to explore how methods such as knowledge graphs and GNN visualization can be integrated to enhance model transparency.

  1. Conclusion

This section is relatively simple and lacks depth. It is recommended to emphasize the potential clinical applications while maintaining appropriate caution. Additionally, it would be beneficial to conclude with a forward-looking statement, highlighting the need for validation in larger sample sizes, multi-center studies, and more in-depth mechanistic investigations.

  1. Materials and methods

The organization of this section appears somewhat disordered, with a lack of tight logical structure. For example, the content under "CircRNA-miRNA prediction and interactions of miRNA-mRNA" should be integrated with "Rationale for Choosing Circular RNA 0009910 and 0027478." Additionally, "The biogenesis and gene structure of circ_0009910 and circ_0027478" would be more appropriately placed within the "Introduction" section. Furthermore, for the prognostic analysis, including survival analysis (e.g., Kaplan–Meier curves) would be preferable; at present, relying solely on diagnostic ROC analysis makes the study's depth appear relatively limited.

Comments on the Quality of English Language

The manuscript contains several grammatical and typographical errors that need to be corrected. For instance, in the sentence "The POLARstar® Omega Nanodrop (BMG Labtech, Germany) was used to assess RNA purity and concentration, ensuring that only high-quality RNA samples with an A260/A280 ratio of at least of 1.8 were used...", the phrase "at least of 1.8" is grammatically incorrect and should be revised to "at least 1.8." Additionally, there are minor errors caused by carelessness, such as an unnecessary period in "To predict probable relationships among circRNAs and miRNAs., the web resources circinteractome (https://circinteractome.nia.nih.gov/) and miRDB (http://www.mirdb.org/) were used (39-41)." Furthermore, in "For every PCR run, the following thermal conditions were used: 95 °C for thirty minutes, 40 cycles of 94 °C for fifteen seconds, 55 °C for 30 seconds, and 70 °C for 30 seconds," "thirty minutes" should be corrected to "thirty seconds."

Author Response

Reviewer 3 Report (New Reviewer)

Comments and Suggestions for Authors

The manuscript titled “Preliminary Evaluation of Plasma circ_0009910, circ_0027478, and miR-1236-3p as Diagnostic and Prognostic Biomarkers in Hepatocellular Carcinoma” by Mona Samy Awed et al. presents an interesting pilot study that investigates the diagnostic and prognostic value of circ_0009910, circ_0027478, and miR-1236-3p in hepatocellular carcinoma (HCC). This research provides novel insights into the roles of circular RNAs and their clinical potential. The authors provide evidence through RT-qPCR and ROC analysis demonstrating circ_0009910's superior diagnostic accuracy (AUC = 0.90), along with significant correlations with tumor size, metastasis, and AFP levels. The integration of logistic regression and correlation analyses further supports the conclusion that these biomarkers and particularly, in combination with AFP may serve as promising tools for HCC detection and monitoring.

In my opinion, these preliminary findings are promising, although the study is limited by its sample size and should be further validated in larger, more diverse cohorts. Overall, the work is well-presented, scientifically sound, and methodologically appropriate, contributing meaningfully to the growing field of circRNA research in oncology. However, the only major concern is that the manuscript does not align closely with the scope of IJMS and may be more appropriately considered for publication in a more clinically oriented journal, such as Current Oncology (MDPI).

Round 2

Reviewer 3 Report (New Reviewer)

Comments and Suggestions for Authors

All my comments were addressed appropriately. However, I still believe this work is better suited for a different journal

Author Response

Reviewer comment: All my comments were addressed appropriately. However, I still believe this work is better suited for a different journal

Reply: We thank the reviewer for acknowledging that all prior comments were addressed appropriately. Regarding the concern that our manuscript may be more suited to another journal, we respectfully submit that the focus on molecular-level expression of circRNAs and miRNA interactions, combined with bioinformatics predictions and expression-based diagnostics, aligns well with the aims and scope of IJMS. Our work investigates non-coding RNA regulatory pathways with clinical translational potential — a topic IJMS has supported in many recent publications. We hope the revised version demonstrates both scientific rigor and thematic suitability for the journal’s readership.

This manuscript is a resubmission of an earlier submission. The following is a list of the peer review reports and author responses from that submission.

Round 1

Reviewer 1 Report

Comments and Suggestions for Authors

The manuscript has significant potential but requires major revisions to improve clarity, rigor, and alignment with the journal standards.  

-       The manuscript has good flow in general, but there is some verbose that need to be removed. In the introduction remove overlapping information on biomarkers. Avoid excessive background on unrelated ncRNAs; focus on circRNAs relevant to HCC. The role or mechanisms of circRNAs in HCC therapy is not relevant and should be minimized. Focus should be on the role of circRNAs as cancer biomarkers and then focus on their role as HCC biomarkers. Also, check repetitions in the discussion.

-        Related to the above, the authors should mention in the introduction that they analyzed the same GSE94508 and GSE97332 analyzed in other studies (Qiu et al. 2019) and found that circ_0027478, circ_0009910, and miR-1236-3p as HCC biomarkers. The authors mention circ_0027478 and circ_0009910 at several locations in the introduction. This is not focused and confuses the readers. These circRNAs were identified through data analysis of GSE94508 and GSE97332 from GEO. But, how did this lead to the decision to study them as HCC biomarkers? It is even less clear why miR-1236-3p was analyzed as HCC biomarker. The relation between miR-1236-3p and the circRNAs circ_0027478 and circ_0009910 is not clear in the introduction?

-       Most references are relevant and cited appropriately. Some critical studies on HCC diagnostics are missing. Add more up-to-date references and cross-check the accuracy of all citations. Add references for specific claims about global HCC prevalence and biomarker limitations. Recent references from 2022–2024 on circRNAs in oncology should be included. Include references to studies validating circRNAs as biomarkers in other cancers. There is a lack of reference of competing biomarkers of HCC and how are circRNA a better HCC biomarker. Include a more robust comparison with other biomarkers. This can be included in the discussion. For example, the authors later mention that AFP is an HCC biomarker. Mention in the introduction that AFP may be used as a biomarker, explain its limitations, and why circRNAs may be a better marker or why there is a need for new HCC markers?

-       The authors should be careful when mentioning "substantially upregulated". Replace with specific statistical values or fold changes.  

-        Use one heading for methods, right now there is a heading at line and a heading at line 197 and one at line 119.

-       In methods, “and the other for AFP activity determination.” Did the authors determine AFP activity or level?

-       The authors need to remove all data related to GSE94508 and GSE97332 from their manuscript. Right now, it is not clear what additional data the authors have obtained that are different from Qiu et al. 2019. Following clarification, only the section on “CircRNA-miRNA prediction and interactions of miRNA-mRNA”  may be retained.

-       Title of the methods “Gene expression analysis with microarray detection” is misleading. The authors did not perform a microarray experiment. Also, the description of the method is misleading. Furthermore, add the proper reference after “Five matched pairs of HCC and adjacent paracancerous liver tissues were included in the GSE94508 dataset, while seven pairs of HCC and matched non-tumor liver tissues were included in the GSE97332 dataset.” Reference 32.  

-       Combine Identification of differentially expressed circRNAs (DECs) and Gene expression analysis with microarray detection” and make sure that “Gene expression analysis with microarray detection” is accurate. Check how to consolidate all methods above, following refinement of what to include from the GSE94508 and GSE97332 datasets. See comments above.

-        Sample size calculation was presented in the methods section, but the result of this calculation is not present in the manuscript?

-        Is the data in Table 3 representing fold change? Fold change with respect to what? If it is fold change compared tocontrol, then control should be set as 1. Clarify?

-       Clarify the patient selection criteria. Specify inclusion and exclusion criteria in one or two sentences.

-       For the discussion, highlight clinical relevance more prominently. Acknowledge limitations, including small sample size and lack of independent validation. Discuss that these markers were not validated in an independent cohort. Discuss potential biases, including differences in age or comorbidities between cases and controls. Discuss potential costs and logistical challenges of incorporating circRNA testing in clinical settings. Suggest validating these biomarkers in larger, multicenter cohorts. Propose studying their mechanistic roles in HCC progression through functional assays.

-       The manuscript needs deep proofreading to eliminate some typos. It is advised that a native English speaker proofread the manuscript. Below are EXAMPLES of English errors and typos, but the whole manuscript should be proofread.

and and 50 age- and gender-matched healthy controls”= Remove duplicate “and.”

subsantially upregulated”= Change to “substantially upregulated.”

"devision"= change to should be "division"in the introduction.

“Recent using of next-generation sequencing (NGS)”= This is an awkward phrase= Change to “The recent use of next-generation sequencing (NGS).”

When citing the figures and tables in the text, then use Figure Table not figure table

“to extract total RNA, 100 μL of plasma was used using the reagent of QIAzol lysis”= Remove the redundancy: “was used using.”= for example= was performed using

clinical duties= clinical practice

“In Table 2 presents a comprehensive overview”= Rewrite as “Table 2 presents a comprehensive overview.”

considering the molecular complexity 56 of this disease is crucial to meeting the demand for new therapeutic approaches.= Considering the molecular complexity of this disease, it is crucial to meet the demand for new therapeutic approaches.= It is better to replace with a more clear sentence

Recent study indicates that deregulation of lncRNA= A recent study

“When comparing HCC patients to control group”= Add “the” before “control group.”

“The findings of the research denotes”= check all Subject-verb agreement issues in the manuscript= Change “denotes” to “denote.”

“lately, there has been significant interest”= Capitalize “Lately” in the beginning of the sentence.

Abbreviations like PTEN, AFP and US are inconsistently defined. Define abbreviations upon first use in the abstract and and independently in the rest of the manuscript.

“RNA samples with pure quality A260/A280 at a minimum of 1.8”= Change to “RNA samples with an A260/A280 ratio of at least 1.8.”

“data is shown using the median”= Change “data represent the median”

“circRNAs are pervasive and essential in modulating cellular processes and illnesses, particularly in cancer”= Improve clarity by simplifying: “circRNAs modulate cellular processes and illnesses, especially cancer.”

“subgroups based on tumor differentiation, size, metastasis, cirrhosis grading, and TNM staging that matched with results of other studies.”= Fix tense to “, matching results of other studies.”

Comments on the Quality of English Language

The manuscript needs deep proofreading. Some sections need to be rewritten.

Reviewer 2 Report

Comments and Suggestions for Authors

The authors here reported the detection of two circRNAs (circ_0009910 and circ_0027478) and one miRNA (miR-1236-3p) as the potential biomarkers in HCC diagnosis. Overall, the conclusion is largely supported by the provided methods and results. More notably, this article is hard to read, mainly due to the lacking of illustrating scheme or the model figure. The following points can be used for the further improvement.

 1. In line 93-95, the author refers to In HCC, miR-1236-3p diminish the PI3K/Akt route by down-regulating PTEN and AFP, which slows the growth and spread of the tumor [27]. As seen, PTEN and miR-1236-3p are the tumor suppressor genes, why miR-1236-3p down-regulates PTEN expression? Please check again.

2. A simple illustration of the biogenesis and gene structure of circ_0009910 and circ_0027478 can be made.

3. The regulatory mode of circ_0009910 and circ_0027478 including the downstream targets can be also illustrated as a figure.

4. Many figures are deformed with low resolution. Please remade them in suitable sizes and high resolution.

5. Is the RT-qPCR product confirmed by sequencing to actually verify the back splicing products of circular RNAs rather than the unprocessed circular RNAs?

6. In Table 1, the primers detecting miR-1236-3p and SNORD68 are missing. Please check again.

7. Some sentences are grammarly incorrect. For instance, in line 624-625, "Indicating that circ_0027478/miR-1236-3p/AFP could be identified as a therapeutic target in HCC patients." Please check again througout all the paragraphs.

Reviewer 3 Report

Comments and Suggestions for Authors

This manuscript investigates the diagnostic and prognostic potential of circ_0009910, circ_0027478, and miR-1236-3p in hepatocellular carcinoma (HCC). While the topic is relevant and the findings are potentially interesting, several areas require improvement, and several major concerns are present.

General comment

 - The manuscript will need extensive editing.

- Wherever possible, avoid labeling people according to their disease (i.e. HCC patients). Instead, use person-first language (patients diagnosed with HCC, or patients with HCC).  Please revise throughout the whole manuscript.

Title

- The title is overly enthusiastic ("Unlocking...New Frontier") and lacks precision. A more appropriate title would be concise and accurately reflect the study's scope.

- As this type of study has a limited sample size, it is recommended that the title include “A preliminary (or pilot) study.”

Abstract

In general, the abstract should concisely state the key findings without excessive detail. Consider reducing the length and focusing on the most impactful results.

 - It is better to start the abstract by talking generally about circular RNA and its relation to microRNAs in HCC, then specify the cause behind the selection of the studied circRNAs and the specified microRNA.

-The phrase "substantially upregulated" (line 28) lacks quantification. Specific p-values or confidence intervals for the findings should be included.

- The claim of "highest diagnostic accuracy (AUC = 0.994)" is exceptionally high and requires strong justification and discussion of potential overfitting.

Introduction

- Lines 65 and 66: The sentence "Circular RNAs (circRNAs) are a subclass of lncRNA molecules" should be revised as the provided reference did not support this elaboration. CircRNAs and lncRNAs are distinct classes of non-coding RNAs with different structural characteristics. While both are types of non-coding RNAs, they are separate categories rather than one being a subclass of the other. Authors must note that circRNAs can be formed from various genomic regions, including exons, introns, or a combination of both, and they have distinct biogenesis mechanisms compared to linear RNAs.

- Lines 76 and 77: “CircRNAs use three primary methods to carry out their biological activities: translating into proteins, communicating with RNA binding proteins (RBPs), …etc. The authors should revise the provided citations. They can look at these comprehensive review articles (e.g. https://doi.org/10.3892/or.2023.8597, https://doi.org/10.1093/jmcb/mjad031).

- Line 79: “which can either inhibit or cause cancer in tumors”. Therre is no need for “in tumors.”

- Lines 109 and 110: “investigate the expression and function,” the authors investigate the expression only in this study. The functional assessment needs further experimental studies. 

Methods

- This section must address potential selection biases in participant recruitment and the potential impact of this bias on the findings. Were participants selected randomly? How were potential confounders controlled for?

- Line 125: “None of them had a history of chronic illnesses,” what about the chronic liver disease?

- Which guidelines do the authors institute follow to diagnose HCC? Please provide supportive reference (s).

- Ultrasonography and other imaging techniques applied for diagnosis should be cited officially.

- Line 143: Did age, gender, and other data need “qualitative exploratory interviews” with patients' relatives?! Why did the authors not obtain these data from the patients themselves or the hospital records?

- From where are the controls selected? What are their inclusion/exclusion criteria?

- Line 147: “and processed” was not clear. What type of processing did the samples subject to?

- Time of sample collection from all participants was not clear. This issue is very important as it could impact the expression of the studied non-coding RNAs.

- Line 154: please provide the date of access to this database.

- Lines 152-186: Did the authors run the written works, or did they derive from the supplied citations? Where are the results of all the mentioned online bioinformatic tools??

-  Lines 182 and 183: “Depending on the results of the bioinformatics analysis, hsa_circ_0009910 and 182 hsa_circ_0027478 were chosen for further investigation due to their possible roles in HCC” Where the results of the running bioinformatic tools listed in details in this section? The yields of each online tool used in this study should be provided.

- Line 185: “as it is predicted to be bound to hsa_circ_0027478,…etc.” Where is the evidence?

- Regarding sample size calculation, the authors did not provide the numerical values of the effect size they applied for sample calculation. The provided reference [47] was running 60 patients as a training group and validated their work in an independent cohort of 218 patients. Why did the authors provide this reference?

- As part of data transparency to facilitate work replications by others, the authors should provide the assay ID and/or the Cat# of each assay used in their lab work.

- Did the authors check the integrity of the extracted RNA?

- Lines 213 and 214: provide the last date of access to these online tools.

- Line 240: the authors should provide the original reference of the Livak method, not citing a secondary reference (ref. 52).

- Did the authors run any type of replicates (either technical or biological) in their PCR runs? This issue is part of quality control measures that should be commented on in this context.

- ELISA section: The authors should write the minimum detection level and the inter- as well as the intra-assay coefficient of the variations for the assessed AFP parameter as part of the quality control measurements for this type of assay. Also, the name of the ELISA set should be provided and cited officially in this context.

- The authors did not mention the lab parameters other than AFP, that were presented in the “Results” section [i.e. aspartate transaminase (AST), alanine transaminase (ALT),  and the presence of hepatitis C virus (HCV) antibodies, international normalized ratio (INR), prothrombin time (PT), albumin and bilirubin], from where they obtain these data?

- There is a need for an explanation of how the authors addressed the potential confounders.

Results

- Line 299: provide the reference for this classification.

- The extremely high AUC value raises concerns about potential overfitting. The authors need to demonstrate that their model generalizes well to unseen data. Techniques to assess and mitigate overfitting should be described.

- Figure 8 and Table 5 have duplicated data.

- Did the authors try to analyze the Prognostic power of the studied parameters collectively?

- The study lacks mechanistic investigation, which limits its overall impact.

Discussion

- The authors should provide study limitations by the end of the discussion.

Comments on the Quality of English Language

This manuscript needs extensive editing to enhance the clarity and data flow. It wasn't easy to follow the authors in their elaborations in many sections.

Reviewer 4 Report

Comments and Suggestions for Authors

Summary

The authors selected circ_0009910, circ_0027478, and miR-1236-3p as biomarkers based on bioinformatics analyses of two datasets (GSE94508 and GSE97332), which included 5 matched pairs of HCC and adjacent paracancerous liver tissues and 7 matched pairs of HCC and non-tumor liver tissues. These biomarkers were identified as differentially expressed between HCC and non-HCC samples. The study further validated these findings in a larger cohort of 150 participants (100 HCC patients and 50 controls) to evaluate their diagnostic and prognostic capabilities. The authors demonstrates that these biomarkers has strong potential for clinical applications in HCC diagnosis and treatment. While the study is well-structured and provides valuable insights, it could benefit from clearer justification for biomarker selection, more mechanistic depth, and additional validation steps to enhance the robustness and clinical relevance of the findings.

Some suggestions

  1. It would be beneficial for the authors to clarify whether other differentially expressed circRNAs or miRNAs were identified but not included in the analysis. Providing more details on the criteria used to prioritize circ_0009910, circ_0027478, and miR-1236-3p, such as fold change thresholds, biological relevance, or novelty, would help clarify their selection. Additionally, briefly addressing potential challenges, such as population-specific variability, variations in online database data, or pre-analytical handling issues, could strengthen the discussion of these biomarkers’ potential as diagnostic tools.
  2. Adding further suggestions for validating the proposed pathways, either experimentally or through additional analyses, could enhance the mechanistic depth of the findings. A brief discussion on how these biomarkers could be integrated into existing diagnostic workflows or complement current practices would also be valuable. Contextualizing the findings by mentioning other emerging biomarkers for HCC could provide a broader perspective and further enrich the discussion.
  3. To improve the transparency of data presentation, the authors might consider overlaying individual data points on the box plots in Figure 2, as done in other figures that display outliers. This adjustment would allow readers to better appreciate the variability and consistency of the data. Additionally, ensuring consistent formatting and style across the text, figures, and tables (particularly regarding capitalization and abbreviations) would enhance the paper's overall readability.
  4. The single ROC curves in Figure 7, together with the correlation and regression analyses, provide valuable insights into the biomarkers’ prognostic power. To further support these findings, incorporating additional analyses such as cross-validation, bootstrapped confidence intervals, or subgroup analyses stratified by tumor stage, metastasis status, or cirrhosis grade could improve the robustness and generalizability of the results. (potential trials: including statistical comparisons between biomarkers, such as DeLong’s test, and considering refinements to predictive models, like stepwise regression or LASSO, could further strengthen the clinical applicability of the study).
  5. While the authors mention that none of the participants had a history of chronic illnesses, explicitly clarifying whether specific chronic liver conditions, such as viral hepatitis, alcoholic liver disease, or non-alcoholic fatty liver disease, were considered and excluded could further reinforce the specificity of the findings to HCC. Providing such details could address potential confounding factors and enhance the study's rigor.
  6. Finally, Table 5 offers valuable evidence for the correlation between circRNAs, miR-1236-3p, and clinical parameters, supporting their potential diagnostic and prognostic roles. Expanding the discussion to explore possible biological mechanisms underlying these correlations and proposing experimental validation could further strengthen the findings. Additionally, explicitly comparing circRNAs with AFP and discussing their complementary or superior utility would highlight the biomarkers’ clinical relevance and potential applicability in HCC management.

Reviewer 5 Report

Comments and Suggestions for Authors

Thank you for the opportunity to review the manuscript by Awed et al. The study aimed to evaluate the diagnostic and prognostic capability of 2 circRNAs in combination with miR-1236-3p when detected in the plasma.

There are grammatical errors throughout. Most notably issues with punctuation, syntax, and formatting.

Lines 52 -54 are misleading. The GALAD index, incorporating the AFP, AFP-L3, and DCP biomarkers, is being increasingly utilized for HCC surveillance in the clinical setting.

Methods. The section on HCC diagnosis and staging is a little confusing. I would recommend concisely focusing on how HCC was diagnosed. It is unclear whether the CT scans were diagnostic (triple phase) imaging, so this would have no impact on diagnosis. All HCC was biopsy confirmed. While unconventional, this would at firmly establish diagnosis. Based on the description of symptoms / indicators for CT imaging, I presume this is not a patient population under active HCC surveillance. This is not going to be the best cohort to evaluate diagnostic potential, as these patients have well established, symptomatic disease.

 Another major study design issue regarding diagnostic sensitivity is the use of matched healthy controls. Healthy controls are not the correct control group for distinguishing HCC. The HCC cohort has 79% HCV history, >80% are Child-Pugh B-C (> 50% Class C), albumin of 2.6 g/dL, bilirubin of 6.8 mg/dL, and an INR of 1.5. I would argue this cohort has well established end-stage liver disease. For diagnostic sensitivity, your control cohort should be patients with end-stage liver disease matched to the metabolic panel and functional status of these patients and with recent imaging without suspicion for HCC and a negative AFP. Also of great concern, about 50% of the HCC cohort are either terminal stage or have severely limited liver functional reserve. Metabolic dysregulation is a hallmark of end-stage liver disease. How will the study be able to confirm circRNA dysregulation is due to the presence and stage of the HCC and characteristic of end-stage liver disease?

The candidate circRNA discovery was performed using tumor / adjacent non-tumor pairs from the GEO. HCC staging for the 5 pairs (GSE97332) and 7 pairs (GSE94508) would be critical information, as would the underlying etiology of liver disease. These public datasets can favor HBV HCC, which develops on a non-cirrhotic background in patients with well-preserved liver function and does not ideally lend itself to modeling toward an HCV HCC population with significant underlying liver disease.

Results. Ln. 276 “substantially older age with no significance”. Why not clearly state the cohort was well controlled for age (P value), gender (P value)… as opposed to estimating the magnitude only to later say it was well controlled.

Table 1. I view this as two tables merged into a single table. The component of the table comparing healthy to HCC-cirrhosis is supplemental material that is not critical to the study. The methods state the cohort was recruited to match for sex and age. A table demonstrating an effective match in supplement would suffice. The bottom portion of the table is the HCC-cirrhosis demographic table, which should be in the main text. Now we have the metabolic panel baseline. I do not see any value in comparing patients with end-stage liver disease to healthy controls. That data should ideally be in the HCC-cirrhosis demographic. If the editor has no reservations, you could include the statistical comparison of the metabolic baseline in the supplement Table.

Some questions regarding Table 3. I initially presumed this was the fold-change of the target normalized to GAPDH, as outlined in the methods. But the Table footnote and text say the data was normalized again to the control group. In this case, I would expect the control median to approach 1.0 with an IQR slightly above and below. I am not sure how normalizing to control results in values < 1.0. Secondly, from my own experience, the use of a single loading control in patients with cirrhosis is quite difficult due to impaired liver function. The expression of traditional housekeeping genes between cirrhosis and healthy controls is often significantly different. This warrants development in the methods section or supplemental data demonstrating the effectiveness of this approach.

The summary data in Table 3 appears in graphical format in Figure 1. The tabular data alone is sufficient.

Figure 2.

Line 320. I would strongly caution against the TMN-based grouping strategy utilized by the authors. Grouping T1-T2 as early stage includes T2 patients with vascular invasion. Under the BCLC system, this would advance stage disease. Unfortunately, the authors cohort does not lend itself well to grouping strategies for staging, as the cohort is dispersed across multiple stages and more weighted toward advance to terminal stage disease. Secondly, the TMN system does not consider underlying liver disease, which highly interacts with tumor biological factors in influencing tumor-dependent outcomes and overall survival outcomes. My recommendation would be to subgroup BCLC A-B vs. BCLC C-D if the authors wish to compare expression levels across HCC staging.

 However, BCLC grouping also presents challenges, especially in a biomarker discovery study. Significant changes in expression level in BCLC-D can be associated with multi-organ failure and less directly to tumor biology, particularly if they are not being measure specifically at the tumor site and controlled for other aspects of aggressive tumor biology.

There are also issues with the figure and legends. The legend describes a multi-summary system including the median, IQR, and 90-10 percentile ranges. However, the figure only shows the median and IQR. Secondly, the variance in these measures is substantial. For instance, while the median values trend for 0009910, the IQR stretches from ~ 5-fold to 100-fold for BCLC B-C, which would make staging sensitivity challenging.

Figure 3. The comparisons based on metastatic vs. non-metastatic are like valid and should mirror any condensed staging-based comparison as metastatic disease indicates major shifts in tumor biology. However, single vs. multifocal disease alone is confounding as this may cross multiple stages of HCC and is not well controlled, especially considering the authors have already highlighted the potential for stage-based changes in expression. Comparisons for tumor size would ideally occur within discrete stages where these influential factors (metastasis, vascular invasion, poor performance status) are already controlled.

Figure 4. Fibrosis staging using traditional ultrasound is unreliable. This data should be removed.

Figure 5. The breakdown of AFP across HCC-linked parameters is well established in the literature and does not advance the aims of the manuscript. This should be supplemental data or excluded as failing to have these anticipated differences would suggest a major problem with study design.

Diagnostic Sensitivity. There is substantial difference between differentiating established HCC from healthy controls and diagnostic sensitivity in patients under HCC surveillance. I do not have an issue with establishing sensitivity versus healthy controls, but any mention of diagnostic accuracy or sensitivity should be removed. This study is not designed to address diagnosis. Similarly, if metastasis is already clinically established via imaging, I am not convinced the interpretation of the data should be as strong as delivered in the manuscript. The clinical prognostic value would be in an associated with developing metastasis prior to detection by imaging. This is established metastatic disease prior to diagnosis which was detectable at the time of diagnosis. In that context, this is more likely an association with overall disease burden and vascular access resulting in higher target spillover into the circulation. Limitations in the study design prevent the authors from answering these critical questions related to prognosis of metastasis.

Line 448, section on predictors associated with risk of diagnosis. The study is not designed to ascertain risk of being diagnosed. An HCC diagnosis is already established, and the control group is not an appropriate reference population for this task.

Graphical Statistics. The authors list several approaches in the methods section, but it is unclear which test were employed for each figure. This information should be included in the figure legends.

Comments on the Quality of English Language

The grammar is problematic throughout and could benefit from an editorial review.

Round 2

Reviewer 2 Report

Comments and Suggestions for Authors

The authors have well answered all my concerns. The resubmitted manuscript is well revised with the much improved quality. This revised manuscript can be accepted in its present form.

Reviewer 3 Report

Comments and Suggestions for Authors

Thanks to the authors for addressing all the raised concerns. Just confirm providing a citation for this sentence before the final publication of the manuscript (line 617): "The circular RNAs were meticulously selected based on their differential expression pattern in HCC compared to non-tumor liver tissues, as identified in the previous study." [Ref?]

Best wishes

Reviewer 5 Report

Comments and Suggestions for Authors

I thank the authors for their responses to my questions and comments. Unfortunately, the responses and revisions do not address my concerns that the circRNA profiles are largely driven by end-stage liver disease. While the circRNAs may be upregulated with respect to healthy controls, they vary substantially across HCC stages, with the highest expression levels in terminal stage disease. The IQR across staging highly overlap. Without control for non-malignant cirrhosis, we cannot link circRNA expression to the presence of HCC or HCC progression with any degree of certainty. My specific response to major concerns that remain and outline below.

In Comment #5, my major concern was the expression level of the circRNA in terminal stage disease (BCLC-D). The author’s do not provide a breakdown of BCLC-D in their cohort, but we know from the staging algorithm that tumor burden is irrelevant in this stage. BCLC-D is characterized by end-stage liver disease and poor performance status. That is why this is such a critical point based on the data. First, we should acknowledge that  Figure 1C and 1D appear to display an ANOVA with repeated t-tests (with no correction for multiple comparisons). This would best be evaluated using a nonparametric test (Kruskal-Wallis) with the appropriate multicomparisons secondary test (Steel-Dwass). More importantly, the authors highlight an association with tumor size (<3cm≥, uncontrolled for staging), metastasis, and AFP (not shown in the data), but dismiss that circRNA have the highest expression levels in BCLC-D (Figure 1C-1D), and that they are only able to detect difference between earlier stage and terminal disease. In this response, the authors claim a role of the circRNAs in oncogenesis, which is not supported by the data as the circRNAs show higher exression level with late-stage disease, which would be consistent with disease progression not disease development (oncogenesis).

Comment 6 (Figure 3) even more clearly demonstrates this strong correlation with the progression of end-stage liver disease and cirrhosis, much more clearly than with HCC progression. This is context of the same statistical test errors in Figure 3 as present in Figure 1.

Comment 11 suggests the authors shifted from the TMN system to BCLC system for comparing early- vs. later-stage, but Figure 1 still says TMN.

Comment 13. The authors are correct that selected figures (Figures 2-3) match the legend labels for summary statistics. However, Figure 1 still does not have the full display.
